# Functional limb muscle innervation prior to cholinergic transmitter specification during early metamorphosis in *Xenopus*

Francois M Lambert[1]*, Laura Cardoit[1], Elric Courty[1], Marion Bougerol[2], Muriel Thoby-Brisson[1], John Simmers[1], Hervé Tostivint[2], Didier Le Ray[1]*

[1]Institut de Neurosciences Cognitives et Intégratives d'Aquitaine, Université de Bordeaux, Bordeaux, France; [2]Evolution des Régulations Endocriniennes, Muséum National d'Histoire Naturelle, Paris, France

**Abstract** In vertebrates, functional motoneurons are defined as differentiated neurons that are connected to a central premotor network and activate peripheral muscle using acetylcholine. Generally, motoneurons and muscles develop simultaneously during embryogenesis. However, during *Xenopus* metamorphosis, developing limb motoneurons must reach their target muscles through the already established larval cholinergic axial neuromuscular system. Here, we demonstrate that at metamorphosis onset, spinal neurons retrogradely labeled from the emerging hindlimbs initially express neither choline acetyltransferase nor vesicular acetylcholine transporter. Nevertheless, they are positive for the motoneuronal transcription factor *Islet*1/2 and exhibit intrinsic and axial locomotor-driven electrophysiological activity. Moreover, the early appendicular motoneurons activate developing limb muscles *via* nicotinic antagonist-resistant, glutamate antagonist-sensitive, neuromuscular synapses. Coincidently, the hindlimb muscles transiently express glutamate, but not nicotinic receptors. Subsequently, both pre- and postsynaptic neuromuscular partners switch definitively to typical cholinergic transmitter signaling. Thus, our results demonstrate a novel context-dependent re-specification of neurotransmitter phenotype during neuromuscular system development.

DOI: https://doi.org/10.7554/eLife.30693.001

*For correspondence:
francois.lambert@u-bordeaux.fr (FML);
didier.le-ray@u-bordeaux.fr (DLR)

**Competing interests:** The authors declare that no competing interests exist.

## Introduction

The features that define a specific neuronal phenotype are generally conserved between species, are specified early during development, but can also undergo adaptive plasticity related to activity after maturation (*Demarque and Spitzer, 2012*; *Borodinsky et al., 2014*). Amongst the large variety of neuronal phenotypes, the developmental properties of the motoneuronal class are particularly well established and are shared by all vertebrates. In mammals, motoneuron (MN) specification begins in the ventral neural tube with the induction of progenitors by sonic hedgehog proteins (*Roelink et al., 1995*), the graded concentration of which triggers the subsequent expression of specific post-mitotic transcription factors (*Goulding, 1998*; *Jessell, 2000*). The homeodomain-containing protein *Islet*1 is the first molecular marker of MN differentiation (*Ericson et al., 1992*) and induces the later expression of homeobox Hb9, which consolidates the motoneuronal phenotype and participates in MN migration and central motor column formation (*Arber et al., 1999*). Soon after their specification, MNs express the two typical proteins associated with cholinergic neurotransmission, choline-acetyltransferase (ChAT) and the vesicular acetylcholine transporter (VAChT), enabling them thereafter to activate their muscle targets (*Phelps et al., 1991*; *Chen and Chiu, 1992*). The muscles, which develop concomitantly and contribute to motoneuron axon path-finding, also play a fundamental role in MN phenotyping and survival (*Yin and Oppenheim, 1992*;

*Kablar and Belliveau, 2005*). Finally, MNs are considered to be fully functional once they have become affiliated to a corresponding central motor network and provide impulse-elicited excitation to muscle fibers using acetylcholine (ACh) as the neurotransmitter (*Davis-Dusenbery et al., 2014*).

In vertebrates, axial MNs innervating trunk muscles are distributed rostro-caudally along the spinal cord in the medial motor column (MMC), whereas fore- and hind-limb MNs are located in the brachial and lumbar lateral motor columns (LMC), respectively. In the amphibian *Xenopus laevis*, the axial and appendicular MNs controlling tail and limb muscles respectively are generated during two separate developmental periods. The former develop during pre-hatchling embryonic stages and control larval undulatory tail-based swimming by projecting to and exciting axial myotomes via nicotinic ACh receptor activation (*van Mier et al., 1985*; *Sharpe and Goldstone, 2000*). Thus, axial neuromuscular ontogeny takes place under conditions equivalent to those in mammals where the MNs and target muscles of a primary motor system develop simultaneously. In contrast, MNs of the neuromuscular system responsible for later limb-based locomotion differentiate and the limb buds start developing during early metamorphosis (*Marsh-Armstrong et al., 2004*), when the axial MNs and tail muscles are already entirely developed and operational.

Given that the primary axial and secondary appendicular MNs of *Xenopus* initially share the same anatomical and neurochemical environment, and because such features influence neuromuscular development (*Yang and Kunes, 2004*; *Menelaou et al., 2015*), it is conceivable that the emergence of the secondary limb MNs and their target muscles is susceptible to influences exerted by the already fully established and functional axial neuromuscular system. We thus hypothesized that the axial myotomes and their innervation constitute a potential interfering environment for the newly developing appendicular neuromuscular system, and that consequently, the latter may have to follow particular developmental rules adapted to this unusual context. In the present study, therefore, we investigated the developmental strategy employed by the early developing limb MNs and associated neuromuscular junctions, with a specific interest in exploring their neurochemical phenotype and functional capability, both in terms of spinal locomotor circuit interactions and limb bud muscle control. Our results show that during a brief pre-metamorphic period, the emerging limb MNs transiently express a non-cholinergic transmitter phenotype that involves glutamate, while exhibiting all other characteristics of typical vertebrate MNs.

## Results

### Spatial organization of the appendicular spinal motor column during early metamorphosis

Appendicular MNs already project into the hindlimb buds as soon as the latter begin to emerge at stage 48 (*van Mier et al., 1985*). At stage 50, the nascent limb consists of a tissue protrusion that is visible on the ventral side of the rostral tail myotomes, next to the larva's abdomen (*Nieuwkoop and Faber, 1956*). The limb bud then continues to grow and differentiate throughout the pre-metamorphic period, during which time the animal triples in size (*Figure 1A₁*). At stage 56/57, the adult-like hindlimb is formed with differentiated thigh and leg segments along with the appearance of five webbed toes (*Figure 1A₂*). The limb extensor and flexor muscle groups are also differentiated at this time.

To investigate the central spatial organization of the motoneurons innervating the hindlimb muscles during this developmental period, we injected two different retrograde tracers into a hindlimb bud and ipsilateral axial myotomes of stage 50 to 57 animals to label the somata and dendrites of appendicular and axial MNs, respectively. Each population is located centrally in separate motor columns. Specifically, the axial motor column is distributed ventro-medially along the entire length of the spinal cord, whereas the hindlimb motor column is located more medio-laterally and restricted to spinal segments 7 to 9, identified by counting caudally the number of ventral roots from the obex (*Figure 1B–D*), as reported previously (*Hulshof et al., 1987*). In early stages 50–52 (*Figure 1B₁, B₂*), retrograde labeling from the limb bud revealed a high density of MNs with relatively small cell bodies (<10 μm) and a reduced dendritic arbor, located more dorso-laterally than the axial MNs. From stage 55 onward (*Figure 1C*) the appendicular motor column is positioned more laterally due to the enlargement of these spinal segments with a lower neuronal density. The appendicular MN somata then become bigger (20–50 μm) and acquire a characteristic elongated

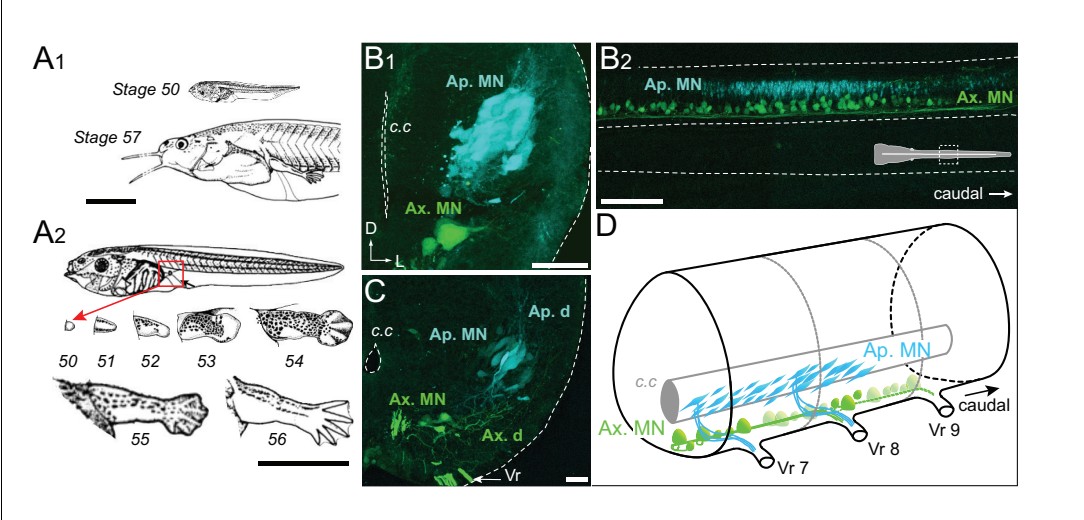

**Figure 1.** Developmental stages of *X. laevis* and anatomical organization of the spinal motor columns. (**A**) Relative size differences between stage 50 and 57 larvae (**A₁**) and associated morphological changes of the hindlimb bud during metamorphosis onset (**A₂**), from **Nieuwkoop and Faber (1956)**. (**B, C**) Segmental (**B₁, C**) and rostrocaudal (**B₂**) organization of the spinal motor column region containing limb MNs at stages 50–52 (**B₁, B₂**) and 55 (**C**). Inset in B₂ shows the location of labeled MNs in the mid-region of the spinal cord. (**D**) Schematic representation of the segmental organization of the appendicular and axial motor columns in the larval spinal cord. Ap./Ax. MNs, appendicular/axial motoneurons; Ap./Ax. d., Ap./Ax. dendrites; Vr, ventral root; c.c, central canal; D, dorsal; L, lateral. Scale bars: A₁ and A₂ = 5 mm; B₁ and C = 20 μm; B₂ = 100 μm.
DOI: https://doi.org/10.7554/eLife.30693.002

cell body shape and extended dendritic arbor, extending from a ventro-medial to dorso-lateral region of the hemicord (**Figure 1C**).

## Delayed cholinergic transmitter phenotype expression in the developing appendicular motor column

The time course of ACh neurochemical ontogeny in appendicular MNs was first investigated using fluorescence immunochemistry for ChAT and VAChT expression in spinal cord cross-sections at different developmental stages (**Figure 2A,B**). At stage 53 (**Figure 2A**), appendicular MNs were not labeled at all with the ChAT and VAChT antibodies, whereas in the same spinal slices, axial MNs were strongly positive for the two proteins. In contrast, at later stages (*e.g.*, stage 57 in **Figure 2B**) both ChAT and VAChT were clearly expressed in appendicular MNs (see lower left insets) as well as in axial MNs. Overall, ChAT and VAChT were not immuno-detected in appendicular MNs until stage 55, in contrast to axial MNs which were immunoreactive throughout the early pre-metamorphic stages.

Semi-quantitative analysis of ChAT/VAChT fluorescence intensity in axial and appendicular MNs (see Materials and methods) was next performed on confocal image stacks in which appendicular MNs were identified by retrograde tracer labeling. The fluorescence variation ($\Delta F/F$) measured at stages 49–54 showed that the immuno-signal detected in the appendicular motor column was not significantly higher than the background signal level (ChAT: $13.5 \pm 5.5$ at stage 49–51; $1.2 \pm 1.0$ at stage 52–54; VAChT: $1.3 \pm 0.8$ at stage 49–51; $5.2 \pm 2.0$ at stage 52–54; **Figure 2C**, right). Immuno-labeling in the LMC became more robust with further larval development, with the appendicular/axial fluorescence variation increasing to ~20% (at stage 55–57; ChAT: $120.4 \pm 31.8$; VAChT: $72.4 \pm 3.9$; **Figure 2C**, right), which was consistent with the clear detection of both ChAT and VAChT in appendicular MNs at the later pre-metamorphic stages. In contrast, the fluorescence variation expressed in axial MNs (**Figure 2C**, left) for both ChAT and VAChT immuno-signal was significantly higher than the background signal level throughout the entire developmental period examined (ChAT: $644.7 \pm 205.4$ at stage 49–51; $849.1 \pm 135.1$ at stage 52–54; $1358.1 \pm 270.9$ at stage 55–57; VAChT: $273.9 \pm 57.5$ at stage 49–51; $267.5 \pm 36.8$ at stage 52–54; $872.5 \pm 134.5$ at stage 55–57; **Figure 2C** left).

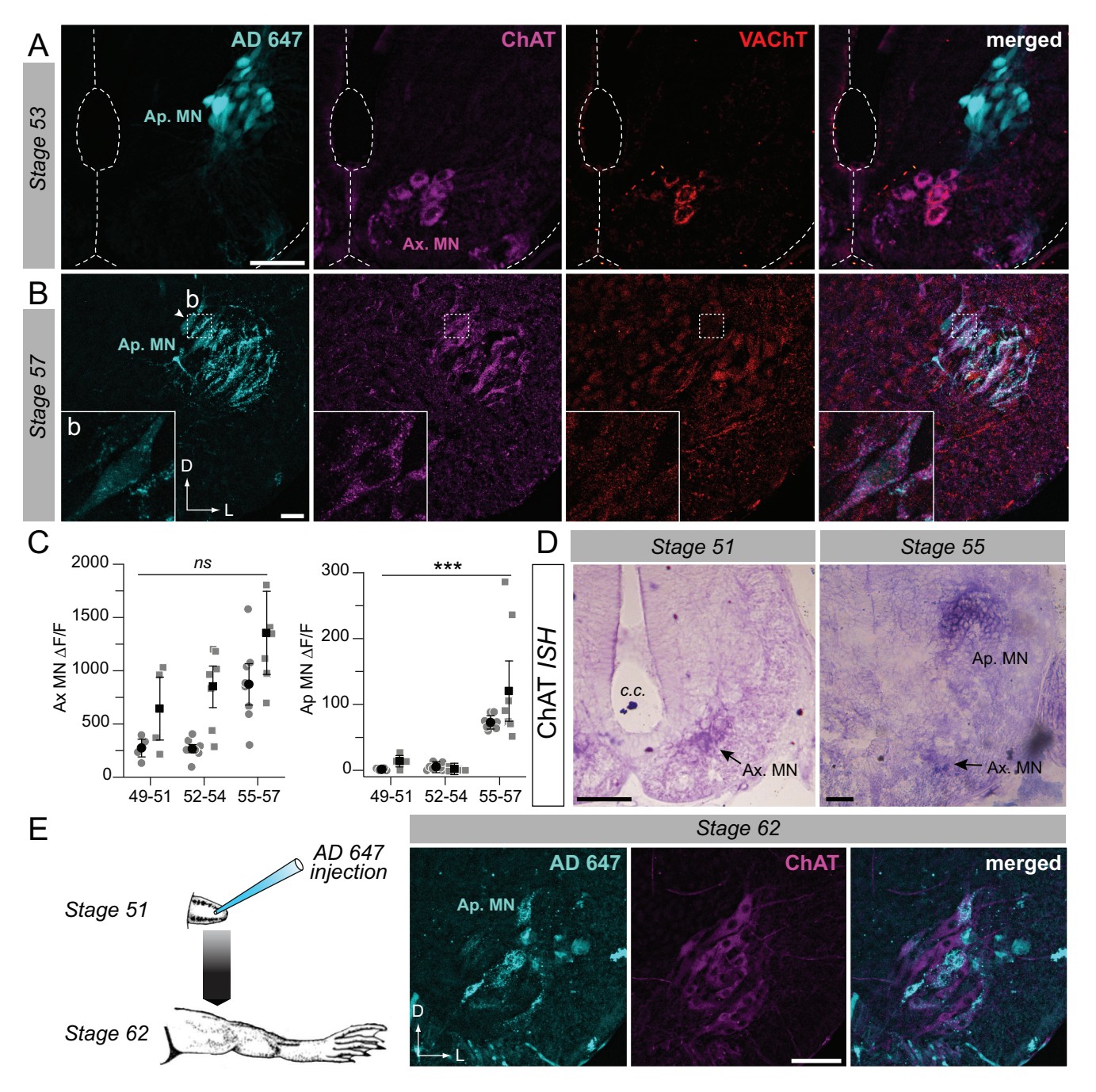

**Figure 2.** Developmental emergence of the molecular ACh phenotype in appendicular MNs. (**A, B**) Examples of fluorescence immunolabeling against ChAT (magenta) and VAChT (red) in appendicular MNs (Ap. MN) labeled with retrograde Alexa Fluor dextran 647 (AD 647, cyan) in stages 53 (**A**) and 57 (**B**) tadpoles. Insets (**b**) in B show x60 magnification of stage 57 appendicular MNs. (**C**) Variation of fluorescence (ΔF/F) of axial (left plot) and appendicular (right plot) MNs for ChAT and VAChT immuno-signals at stages 49–51 (n = 4), stages 52–54 (n = 7) and stage 55–57 (n = 8). Grey dots represent the averaged ΔF/F values for ChAT (squares) and VAChT (circles) in each preparation, and the black dots are ΔF/F grand means ± SEM for all preparations in a given developmental group. *ns* non significant, ***p<0.001, Kruskall-Wallis test. (**D**) Examples of *in situ* hybridization (*ISH*) labeling for ChAT mRNA in the appendicular spinal column at stages 51 and 55. (**E**) Example of fluorescence immunolabeling against ChAT in stage 62 appendicular MNs that were previously labeled with AD 647 injected into the hindlimb at stage 51 (see schematic at left). All scale bars = 50 µm; D, dorsal; L, lateral.

DOI: https://doi.org/10.7554/eLife.30693.003

*In situ* hybridization (*ISH*; n = 16) was performed for ChAT mRNA detection in a larval group ranging from stages 49 to 57. No signal was detected in the appendicular motor column area in stages 49–52 (*Figure 2D* left; n = 7). A ChAT *ISH* signal was then weakly detectable from stage 53–54 (n = 2), to become strongly evident from stage 55 onwards (*Figure 2D* right; n = 7). In contrast, a strong ChAT *ISH* signal was observed in axial MNs throughout the entire developmental period examined. Because *ISH* detects mRNAs that precede protein synthesis, our immunochemistry and *ISH* results are consistent and together confirm that appendicular MNs do not exhibit the cholinergic molecular phenotype prior to stage 55, although they innervate the hindlimb bud muscles from stage 48.

One possibility was that the non-cholinergic and cholinergic appendicular MNs observed at different developmental stages were in fact distinct populations. To address this possibility we first applied a fluorescent retrograde dye to the hindlimb bud at stage 51/52, prior to the appearance of the ACh phenotype in limb MNs (*Figure 2E*). Thereafter, the tracer-treated larvae were raised until they reached metamorphosis climax, by which time the appendicular system is presumably fully developed. ChAT immunolabeling then performed on spinal cross sections taken from these stage 62 animals strongly labeled the appendicular motor population (*Figure 2E*, middle panel). Significantly, some of these ChAT-positive MNs were also co-labeled with the retrograde fluorescent dye previously applied at earlier stage 51/52 (*Figure 2E*, right panel). Thus, these double-labeled MNs that innervated the hindlimb at stage 62 were already present at stage 52, before they expressed the cholinergic molecular phenotype. This finding therefore demonstrated that early non-ACh appendicular MNs constitute at least a sub-population of the cholinergic MNs that constitute the future mature appendicular motor innervation after metamorphosis.

## Molecular and functional identification of early non-cholinergic appendicular neurons

The molecular identity of the non-ACh neurons innervating the newborn limbs at early pre-metamorphic stages was verified by immunochemistry against the transcription factor *Islet*1/2, a protein marker of motor neurons (*Ericson et al., 1992*). Double immunostaining in larvae younger than stage 55 revealed that the ChAT-negative neurons labeled with retrograde dye applied in the hindlimb bud were also strongly *Islet*1/2-positive, similar to the mature axial MNs present in the same slices (*Figure 3A*). Comparable results were obtained with *ISH* for *Islet*1 (*Figure 3—figure supplement 1*).

The electrophysiological properties of these non-ACh appendicular MNs were then tested at stage 52 by performing patch-clamp intracellular recordings of neurons identified by RDA retrograde labeling from the hindlimb bud (*Figure 3B*). These recorded MNs had average resting membrane potential values of $-40.2 \pm 8.2$ mV (n = 12) and $-54.8 \pm 2.3$ mV (n = 5) with low and high [Cl$^-$] patch solutions, respectively, and were able to produce action potentials either in response to increasing steps of depolarizing current injection (*Figure 3C*, left), on rebound after release from experimental hyperpolarization (*Figure 3C*, right) or even spontaneously (*Figure 3D*). Significantly, moreover, during spontaneous episodes of locomotor-like activity monitored from a more caudal spinal ventral root, correlated synaptic fluctuations were observed in all identified appendicular MNs, thereby indicating a functional synaptic coupling with other components of the spinal locomotor network (*Figure 3E*). In addition, the use of an elevated [Cl$^-$] solution in the intracellular electrode (see Materials and methods) revealed a capability for locomotor-related action potential firing in all recorded MNs (*Figure 3F*), and was consistent with the idea that chloride-mediated signaling was still not fully mature in limb MNs and remained excitatory at early developmental stages (*Hanson and Landmesser, 2003*; *Akerman and Cline, 2006*).

Altogether, these findings demonstrate that the retrogradely-labeled, non-ACh spinal neurons that prematurely innervate hindlimb buds express the motoneuronal marker *Islet*1/2 and present basic biophysical characteristics of functional MNs, being able to produce impulses as a function of membrane potential that in turn can be influenced via synaptic influences from central premotor circuitry.

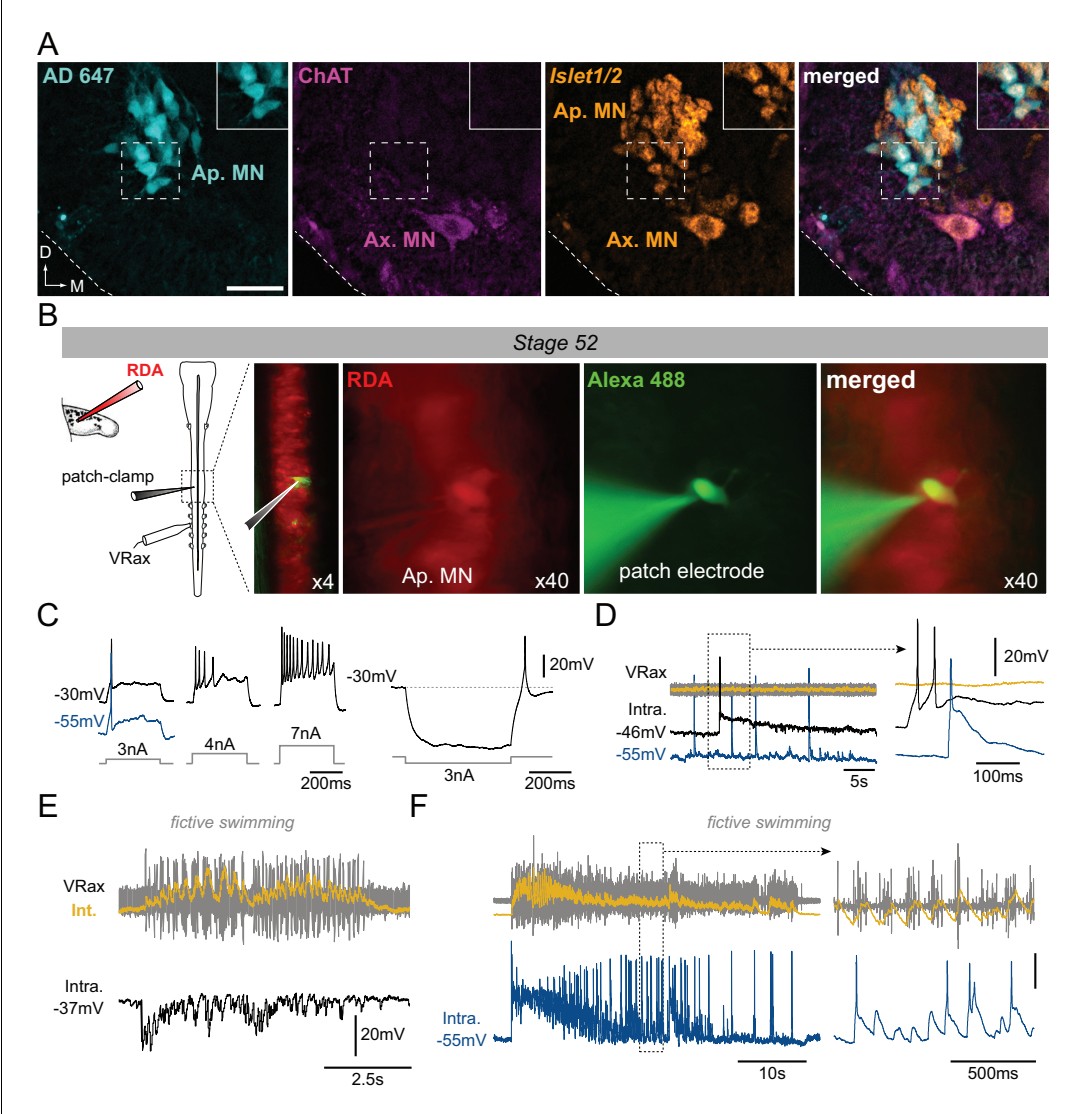

**Figure 3.** Motoneuronal identification of non-cholinergic limb projecting neurons. (**A**) Example of fluorescence immunolabeling against ChAT (magenta) and *Islet*1/2 (orange) in appendicular and axial MNs (Ap. MN, Ax. MN) in a stage 52 tadpole. Appendicular MNs were previously labeled with retrograde Alexa Fluor dextran 647 (AD 647, cyan). Scale bar = 50 µm; D, dorsal; M, medial. (**B**) Protocol for intracellular recordings from appendicular MNs. Stage 52 MNs were identified by prior rhodamine dextran amine (RDA) retrograde labeling from the hindlimb bud and recorded with cell-attached patch-clamp in whole CNS preparations in order to preserve spinal locomotor circuitry. Alexa Fluor 488 added into the recording pipette allowed verifying the motoneuronal identity of the recorded cell. Recordings were made with either a low (black traces in C-F) or high [Cl⁻] (blue traces in C-F) intra-pipette solution. (**C**) Increasing steps of injected depolarizing current elicited increasing spike discharge whereas release from hyperpolarizing current injection evoked rebound spiking. (**D**) Spontaneous MN (Intra) firing in the absence of axial ventral root (VRax) activity. Extended time scale is shown at right. (**E-F**) MNs received strong rhythmic synaptic input during spontaneous episodes of fictive axial locomotion, which triggered locomotor-related spiking in high [Cl⁻]-recorded MNs (**F**). Extended time scale is shown at right. VRax traces (in yellow) are integrated transforms (Int.) of raw extracellular VR recordings.

DOI: https://doi.org/10.7554/eLife.30693.004

The following figure supplement is available for figure 3:

**Figure supplement 1.** Presence of spinal *Islet*1 and ChAT mRNAs at different developmental stages.

DOI: https://doi.org/10.7554/eLife.30693.005

# Locomotor-related activation of hindlimb muscles by non-cholinergic motoneurons

Whether the early appendicular MNs are actually capable of driving muscle activation in the hindlimb buds before the emergence of the cholinergic phenotype was next investigated by making EMG recordings from both limb bud muscles and axial myotomes in semi-intact larval preparations (*Figure 4A* left panel). During episodes of spontaneous axial fictive swimming in a stage 53 preparation (n = 7; *Figure 4A*, right panel), locomotor commands monitored from a caudal spinal ventral root elicited rhythmic EMG activity in both the segmental myotomes and limb bud muscles. The ventral root bursts occurred in phase with EMG bursts in the recorded ipsilateral hindlimb muscle and in alternation with bursts in the contralateral myotome (*Figure 4B*, control). Under subsequent bath application of *d*-tubocurarine to block any nicotinic receptors and thereby prevent ACh-dependent synaptic transmission (*Sillar and Roberts, 1992*; *1993*), the expression of swimming-related ventral root burst activity persisted, but associated tail myotome EMG activity was completely abolished

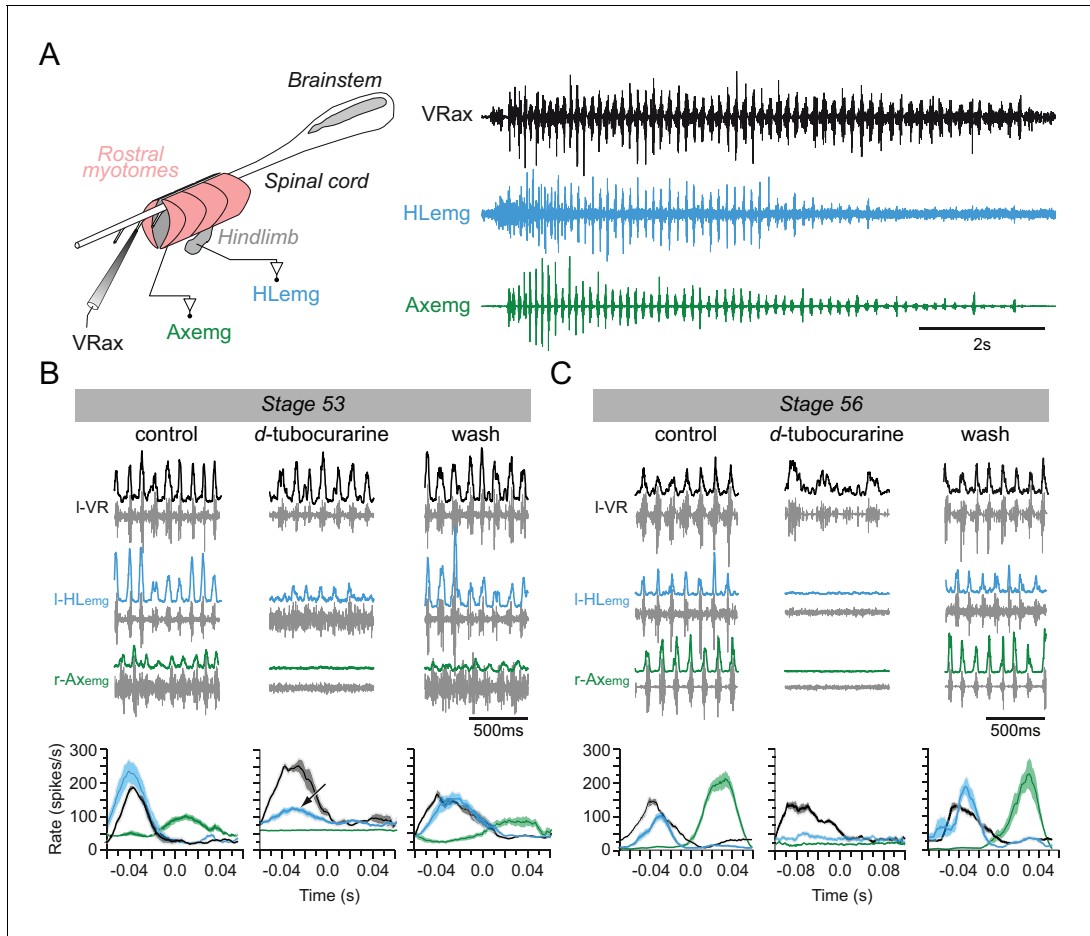

**Figure 4.** Switch from non-cholinergic to cholinergic limb muscle activation during axial swimming at different developmental stages. (A) Rhythmic burst discharge (right panel) recorded from an axial ventral root (VRax) together with a hindlimb bud muscle (HLemg) and axial myotome (Axemg) during a spontaneous fictive swim episode in a semi-isolated stage 53 preparation (left panel). (B-C) Examples of recordings before (control), during and after (wash) bath application of *d*-tubocurarine to a stage 53 (B) and a stage 56 (C) preparation. Lower plots in B and C show instantaneous l-HLemg (blue) and r-Axemg (green) discharge rates (spikes/s) averaged (±SEM) over 10–20 l-VRax locomotor cycles (black). Black arrow in the middle graph of B indicates persistent limb bud EMG activity occurring in phase with the ipsilateral ventral root during *d*-tubocurarine application, and which was no longer present at the later developmental stage (c.f., middle plot in C).

DOI: https://doi.org/10.7554/eLife.30693.006

The following figure supplement is available for figure 4:

**Figure supplement 1.** Appendicular MNs are activated centrally by cholinergic inputs.
DOI: https://doi.org/10.7554/eLife.30693.007

after 15 min of antagonist perfusion (*Figure 4B,d*-tubocurarine). In contrast, even after >30 min *d*-tubocurarine perfusion the hindlimb bud muscles continued to exhibit rhythmic EMG discharges that, although reduced in amplitude (see below), remained strictly coordinated with the ongoing axial pattern (*Figure 4B,d*-tubocurarine; see black arrow in lower middle plot). The ability of both the tail myotomes and limb bud muscles to express locomotor-related activity recovered fully after 2 hr washout with normal saline (*Figure 4B*, wash). Semi-intact preparations older than stage 54 similarly exhibited EMG burst activity in their hindlimb buds and tail myotomes during fictive axial locomotion in control conditions (n = 5; e.g., stage 56 in *Figure 4C*, left). In these cases, however, a 15 min bath application of *d*-tubocurarine decreased the frequency of the centrally-generated axial swim pattern, and completely abolished, albeit reversibly, associated EMG activity in both the hindlimb bud muscles and tail myotomes (*Figure 4C,d*-tubocurarine; wash).

The substantial reduction in limb EMG activity by *d*-tubocurarine at developmental stages earlier than 55 (see *Figure 4B*) could have been due to a peripheral influence on transmission at the neuromuscular junction and/or result from an upstream action on the central activation of the limb motoneurons themselves. To distinguish between these two possibilities, we used immunolabeling and calcium imaging on completely isolated CNS preparations to assess whether functional cholinergic synapses are present on limb MNs within the spinal cord and are directly affected by nicotinic antagonist perfusion. Co-localized sites of VAChT and synapsin labeling were found on the somata of identified limb MNs (*Figure 4—figure supplement 1A*), which also expressed calcium fluorescence changes associated with bouts of fictive swimming (*Figure 4—figure supplement 1B*). Significantly, moreover, the amplitudes of these calcium signals were drastically reduced in the added presence of *d*-tubocurarine (*Figure 4—figure supplement 1C,D*). These findings thus indicated that the reduction in EMG activity observed in semi-isolated preparations (as seen in *Figure 4B*) might not have resulted from a peripheral action of the antagonist at the level of the neuromuscular junction (NMJ) itself, but rather, was largely a consequence of the antagonist's central nervous effects on actual limb MN activation. In this case, the unidentified premotor source of rhythmic drive to limb motoneurons can be presumed to use cholinergic signaling (*Figure 4—figure supplement 1*), although as suggested by the unmasking of depolarizing synaptic events by high electrode chloride concentrations (*Figure 3F*) and earlier findings that *d*-tubocurarine can block GABA-A receptors (*Bixby and Spitzer, 1984*), it is possible that GABA neurotransmission is also involved. The differential effects of *d*-tubocurarine at early and later larval stages also suggested that a developmental shift in the mechanism by which the appendicular MNs activate their target muscles during fictive locomotion occurs around intervening stages 54–55. Whereas neuromuscular transmission appears to be initially independent of ACh signaling in the pre-metamorphic tadpole, it evidently changes to a completely ACh-dependent process in older tadpoles. This in turn suggests that developmental modifications in parallel with the switch in MN transmitter signaling must also occur in the receptor phenotype of the hindlimb muscles themselves.

## Developmental switch in hindlimb neuromuscular transmission

In the *Xenopus* embryo, immature axial MNs have been shown to co-express ACh and glutamate, although glutamate is reported to have no postsynaptic bioelectrical effects (*Fu et al., 1998*). Since our present data show that the appendicular MNs are able to activate hindlimb muscles before using ACh as a neurotransmitter, we asked whether glutamate may be involved in the early NMJ transmission process.

First, we explored in whole-mount hindlimb buds at various developmental stages (from 51 to 57) the putative expression and relative distribution of nicotinic and glutamate receptors with fluorescent α-bungarotoxin and the anti-NR2b antibodies, respectively. At stage 52, there was a total lack of labeling for α-bungarotoxin in hindlimb buds (*Figure 5A*, upper panels), indicating an absence of muscle ACh receptors at this stage. In contrast, at the same developmental stage, diffuse NR2b fluorescence, indicative of glutamate receptor labeling, was observed in the vicinity of MN axons visualized with neurofilament immuno-detection. A similar expression of NR1, another glutamate receptor subunit, was found which also paralleled synaptophysin labeling (*Figure 5B*). In addition, at stage 52, both the axons (*Figure 5C*) and cell bodies of limb MNs (*Figure 5—figure supplement 1A*) strongly expressed the vesicular glutamate transporter VGluT1, which again predominantly colocalized with synaptophysin and in the vicinity of postsynaptic NR1 labeling (*Figure 5—figure supplement 1B1*). Thus, in early pre-metamorphic development, vesicular glutamate transporters are

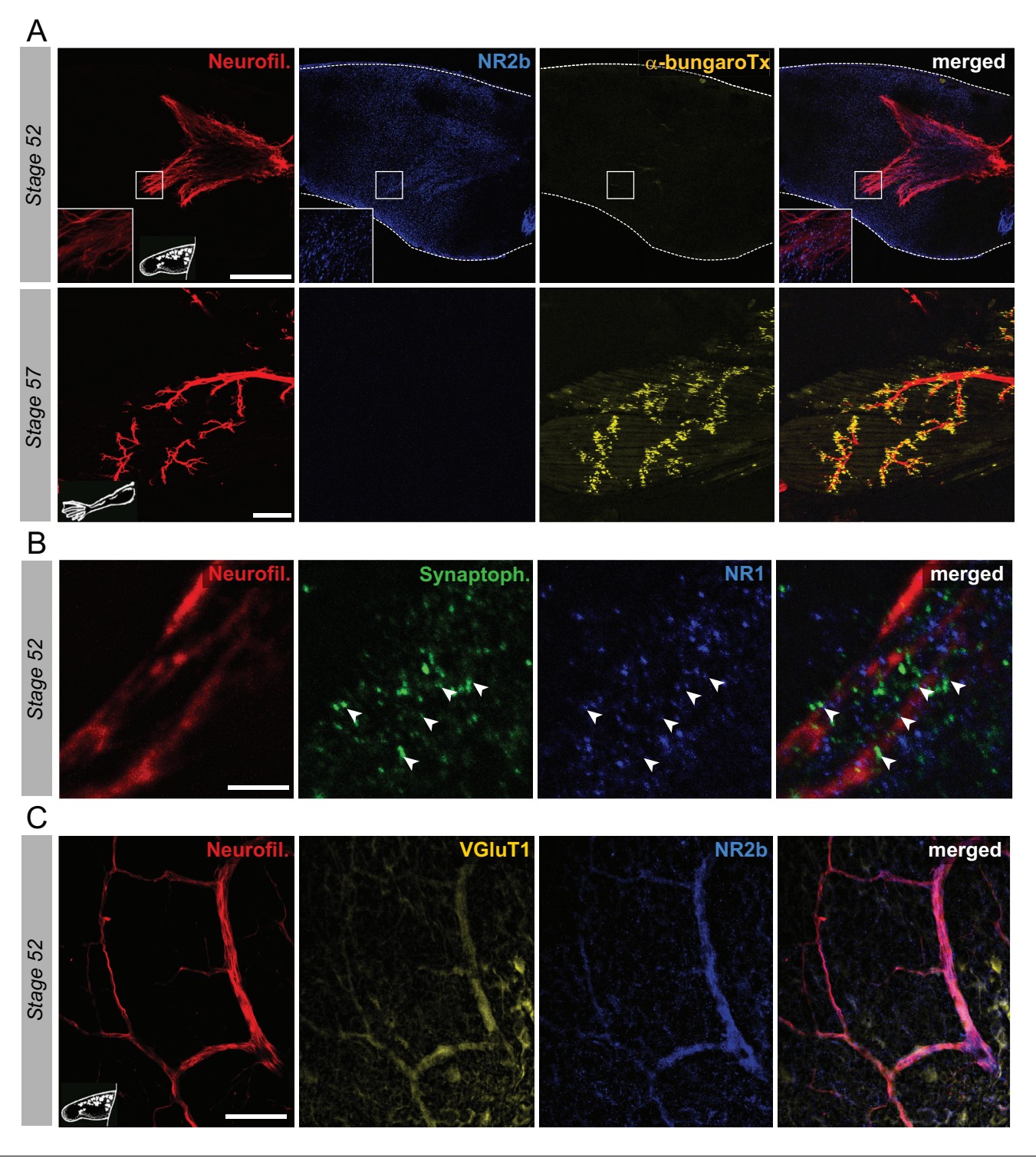

**Figure 5.** Switch from glutamate to acetylcholine receptors in hindlimb muscles. (**A**) Examples of hindlimb innervation patterns and distribution of ACh nicotinic receptors in a whole-mount hindlimb bud, revealed by fluorescence immunolabeling of neurofilament associated protein (Neurofil., red), glutamate receptor (NR2b, blue) and α-bungarotoxin labeling (α-bungaroTx, yellow) respectively, at stages 52 (upper panels) and 57 (lower panels). Inset drawings at bottom left of each panel show bud morphology at the two representative larval stages. Scale bars = 200 µm for stage 52, 100 µm for stage 57. (**B**) Examples of fluorescence immunolabeling against neurofilament associated protein (red), synaptophysin (Synaptoph., green) and NMDA glutamate receptor subunit 1 (NR1, blue) in whole-mount limb bud at stage 52. White arrowheads indicate sites of apposition of all three markers. Scale

*Figure 5 continued on next page*

*Figure 5 continued*

bar = 20 μm. (**C**) Examples of fluorescence immunolabeling against neurofilament associated protein (red), the glutamate vesicular transporter 1 (VGluT1, yellow) and the glutamate receptor subunit NR2b (blue) in whole-mount limb at stage 52. Scale bar = 50 μm.

DOI: https://doi.org/10.7554/eLife.30693.008

The following figure supplement is available for figure 5:

**Figure supplement 1.** VGluT1 expression appendicular MNs.

DOI: https://doi.org/10.7554/eLife.30693.009

present in hindlimb MNs, and concomitantly, glutamate receptors are expressed on hindlimb muscle fibers, close to synaptophysin-marked presynaptic terminals, consistent with the existence of functional glutamatergic NMJs.

The first α-bungarotoxin positive signal was detected at stage 55 with characteristic dispersed dots (data not shown), as previously described (*Marsh-Armstrong et al., 2004*). By stage 57, typical clusters of nicotinic receptors were observed close to appendicular nerve terminal branches, and in further direct contrast to younger stages, NR2b (*Figure 5A*, lower panels) and NR1 receptor subunits (*Figure 5—figure supplement 1B2*) in hindlimb bud muscle as well as VGluT1 in innervating motor axons (*Figure 5—figure supplement 1B2*) were now absent. These findings therefore indicate that glutamate receptor subunits are no longer expressed in hindlimb muscle (and at least one vesicular transporter in motor axons) from stage 55 onwards, but rather, are superseded by nicotinic receptors in likely correspondence to the switch in NMJ transmission to a cholinergic phenotype.

In a final step, the electrophysiological response properties of neuromuscular transmission were tested on isolated hindlimb preparations over a similar range of developmental stages (from 52 to

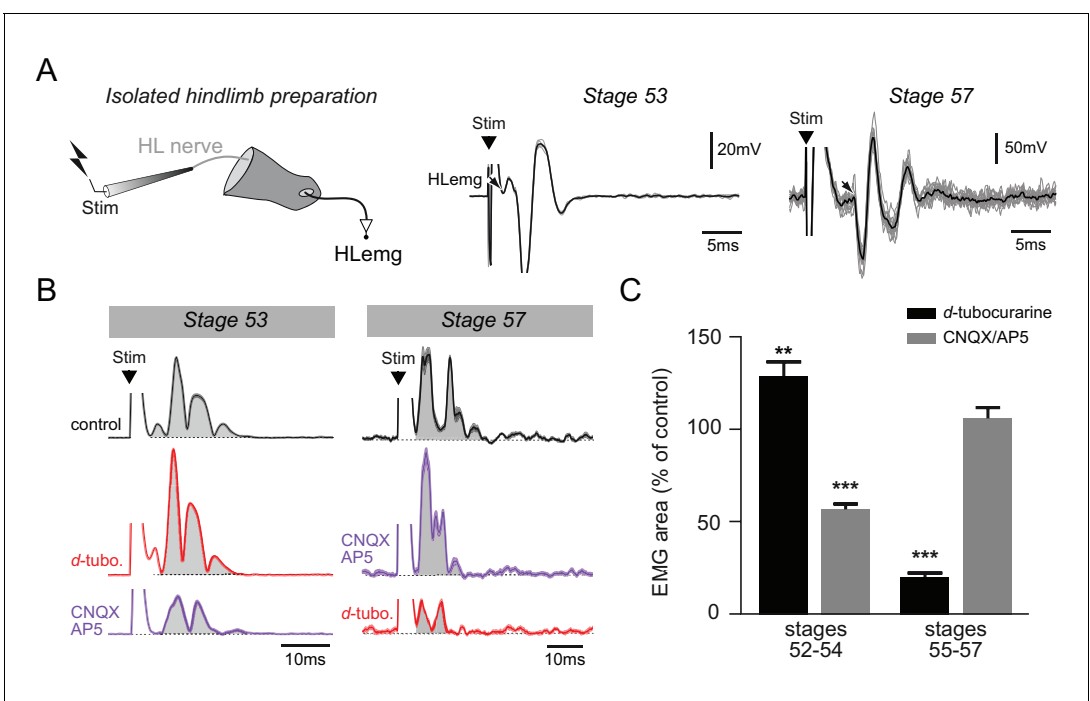

**Figure 6.** Functional switch in hindlimb neuromuscular transmission. (**A**) EMG recordings (HLemg) from an isolated hindlimb preparation (schematic at left) in response to single pulse (10 μs) electrical stimulation (Stim) of a limb motor nerve (HL nerve) in stages 53 and 57 larvae. In each case, the black arrow indicates the beginning of the EMG response and the black trace represents the mean profile of 6 superimposed responses. Note that the trace illustration for stage 53 was taken from a 1 mM Mg²⁺ saline experiment. (**B**) Integrated motor nerve-evoked HLemg responses in control (black), and under *d*-tubocurarine (*d*-tubo.; red) or CNQX + AP5 (purple) bath application to stage 53 and 57 preparations. Thick lines represent the mean response profile (± SEM); the area under each curve (grey) was used to measure the response size. (**C**) Mean (± SEM) EMG response area as percentage of control response during *d*-tubocurarine (black) or CNQX/AP5 (grey) bath application at stages 52–54 and 55–57, respectively. **p<0.01 and ***p<0.001, Mann–Whitney U-test.

DOI: https://doi.org/10.7554/eLife.30693.010

57; *Figure 6*). Single pulse electrical stimulation applied to a limb bud motor nerve elicited EMG responses in the hindlimb bud with a mean latency of 7.8 ± 1.9 ms at stages 52–54 (n = 6) and 5.1 ± 2.1 ms at stages 55–57 (n = 6), and a mean amplitude of 0.30 ± 0.01 mV at stages 52–54 and 2.50 ± 0.99 mV at stages 55–57 in normal Ringer's saline. Either *d*-tubocurarine or CNQX/AP5 cocktail was then bath-applied to test whether these neuromuscular responses were ACh- or glutamate-mediated, respectively. At stages 52–54, the appendicular nerve stimulus-evoked EMG response was not diminished by subsequent *d*-tubocurarine application, but rather, was increased by 35 ± 6% (*Figure 6B*, left; Fig; 6C, left black bar, p>0.01, U = 642.0). [Note that such a *d*-tubocurarine-induced enhancement of post-junctional responses has already been reported in other animal models and is unrelated to its antagonistic effects on nicotinic receptors themselves (*Egan et al., 1993*; *Baron et al., 1996*)]. Conversely, the hindlimb muscle response was decreased by 44 ± 3% following bath-application of CNQX/AP5 (*Figure 6B*, left; *Figure 6C*, left grey bar, p<0.001, U = 39.5). At stages 55–57 on the other hand, the nerve stimulus-evoked EMG response was decreased by 80 ± 2% under *d*-tubocurarine perfusion (p<0.001, U = 74.0) but was not significantly affected (p=0.5, U = 522.0) in the presence of CNQX/AP5 (*Figure 6B*, right; *Figure 6C*, right black and grey bars, respectively).

Altogether, these immunochemical and pharmacological results showed that glutamate, but not ACh, is initially involved in hindlimb muscle activation, whereas NMJ transmission becomes ACh-mediated and increasingly efficient (increase in response amplitude, reduction in transmission delay) from stage 55 onwards. This therefore confirms the occurrence of a functional switch in *Xenopus* hindlimb muscle properties that is commensurate with the switch in appendicular MN neurotransmitter phenotype during the pre-climax phase of metamorphosis.

## Discussion

In this study, we report that as early as pre-metamorphosis (i.e., prior to stage 55), de novo appendicular MNs express the *Islet* transcription factor, exhibit characteristic MN electrical properties and synaptic influences, are involved in locomotor bouts of activity, and project to limb bud muscles to evoke typical post-junctional responses. Unexpectedly, however, these MNs do not at this premature stage use standard cholinergic neurotransmission to activate their target muscles, but it is not until stage 55 that the limb MNs and their NMJs undergo a simultaneous switch from non-cholinergic to acetylcholine-dependent signaling. Thus our data show for the first time that in metamorphosing *Xenopus*, MNs that innervate the newly emerging hindlimbs first develop through the employment of a transient, but functional, alternative transmitter mechanism before a conventional and definitive cholinergic phenotype appears.

Classically in vertebrates, developing MNs are perpetually cholinergic and extend their axons out of the spinal cord towards target muscles to form functional NMJs (*Phelps et al., 1991*; *Goulding, 1998*). However, in *Xenopus* the ontogeny of appendicular MNs does not obey this well-established developmental pattern, since the present data indicate that before the acquisition of a cholinergic phenotype, these MNs are able to transmit centrally-generated motor commands to the developing limb bud muscles using a different transmitter effector(s). Several lines of evidence indicate that glutamate plays a major role in this precursor signaling at early metamorphic stages. Immunolabeling of stage 52 preparations revealed the presence of the vesicular glutamate transporter VGluT1 in identified (retrogradely-labeled) hindlimb-bud MNs, both in their cell bodies, as has been reported in mammalian CNS neurons (*Nakamura et al., 2005*; *Yang et al., 2014*), and peripheral axons (*Melo et al., 2013*). Significantly, we found the axonal VGluT1 to be co-localized with the synaptic protein, synaptophysin, indicative of a presence at the actual NMJ presynaptic terminals.

Also essential to synaptic signaling is the expression of appropriate receptors at the post-junctional level in correspondence with the type of pre-synaptic neurotransmitter released. Accordingly, we observed that the use of glutamate prior to ACh as a peripheral neurotransmitter is matched by the presence of glutamate receptors prior to the appearance of cholinergic receptors at the limb NMJs. Such a parallel developmental sequence is therefore consistent with a neurotransmitter phenotype switching, whereby intrinsic molecular processes lead to a change of one (or several) transmitter(s) to another within the same presynaptic neuron and a concomitant modification in associated postsynaptic receptor subtype in order to ensure synapse functionality. This loss/gain in pre-synaptic neurotransmitter/post-synaptic receptor phenotype has been reported to occur in

developmental, post-lesional or activity-dependent contexts and is considered to be a major under-lying feature of neuronal plasticity (*Spitzer, 2017*). It can either maintain or invert synaptic sign (*Borodinsky et al., 2004*; *Borodinsky and Spitzer, 2007*) and is generally associated with observ-able behavioral changes (*Sillar et al., 1998*; *Demarque and Spitzer, 2010*).

Our immunohistochemical evidence for an early role of glutamate at the developing limb NMJ was also supported by electrophysiological data on the effects of CNQX/AP5 on stimulus-evoked EMG responses in isolated motor nerve/limb muscle preparations. In direct contrast to older, post-stage 55 preparations where these glutamate antagonists were without any effect on evoked muscle potentials, at younger stage 52–54, EMG amplitudes in response to nerve electrical stimulation were strongly reduced. However, that this blockade remained only partial (ca. 50%, in the presence of CNQX/AP5 alone or in combination with the cholinergic receptor antagonist *d*-tubocurarine) might have been due to the low concentrations of glutamate ionotropic receptor antagonists used in our experiments (cf., *Dale and Roberts, 1985*) or that the antagonists were not fully effective on the still immature limb neuromuscular system. Alternatively, metabotropic glutamatergic receptors may be present (*Pinard et al., 2003*) and contribute to neurotransmission at the early developing limb NMJ or (an)other, as yet unidentified, neurotransmitter(s) may be involved.

In *Xenopus*, limb MNs are born during pre-metamorphosis, then their number diminishes after the establishment of NMJs (*Hughes, 1961*; *Prestige, 1967*). This in turn raises the possibility that the initial non-cholinergic motoneuron population we identify here constitutes a short-lived subclass of LMC neurons that project transiently into the limb buds at pre-metamorphic stages, but then is totally replaced by a different cholinergic population as metamorphosis proceeds. However, our finding that limb bud MNs previously retrogradely-labeled at stage 51 are still present at stage 62 when the population of appendicular MNs is fully established and uniformly cholinergic (*Baldwin et al., 1988*), including our early-labeled neurons, argues against this hypothesis. The pos-sibility that the initial non-cholinergic phenotype may be restricted to a specialized motoneuronal sub-population that innervates limb muscle spindles is also unlikely since frog spindles are mainly innervated by collateral branches of skeletal MNs (*Katz, 1949*; *Gray, 1957*). Although a specific fusi-motor innervation has been reported in the bull frog (*Fujitsuka et al., 1987*), since these MNs appear to be restricted to the *semitendinosus* muscle only, they would constitute a very discrete motor subset, the small proportion of which does not correspond to the large number of early non-cholinergic LMC neurons that we find in pre-metamorphic *Xenopus*. Thus, our data strongly indicate that the same motoneuronal population in the developing appendicular motor system of *Xenopus* does indeed utilize consecutively two different neurotransmitter signaling mechanisms in association with the emergence of hindlimb motility (*Hughes and Prestige, 1967*) and forthcoming limb-based locomotor and postural control (*Combes et al., 2004*; *Beyeler et al., 2008*).

Given that limb MNs acquire their cholinergic phenotype around stage 55 and remain cholinergic throughout adulthood (*Baldwin et al., 1988*), the existence of an early, brief but functional non-cho-linergic phenotype is at first sight puzzling. In a broader context, a number of studies have reported the co-existence of various neurotransmitters in vertebrate MNs. For instance, axial MNs of *Xenopus* embryos co-express glutamate and ACh (*Fu et al., 1998*), while developing myotome fibers initially express a variety of receptors, including both cholinergic and glutamate subtypes, until NMJ forma-tion when solely cholinergic receptors are preserved (*Borodinsky and Spitzer, 2007*). Co-released glutamate regulates the development and function of cholinergic neuromuscular synapses in larval zebrafish (*Todd et al., 2004*) and *Xenopus* (*Fu et al., 1998*) by potentiating ACh release through an activation of presynaptic receptors on the MN terminals themselves. Moreover, mammalian spinal MNs co-release glutamate and ACh centrally to activate Renshaw cells (*Mentis et al., 2005*; *Nishimaru et al., 2005*; *Lamotte d'Incamps and Ascher, 2008*), as well as at their peripheral termi-nals (*Waerhaug and Ottersen, 1993*; *Rinholm et al., 2007*) where post-synaptic muscle fibers pos-sess both ACh and glutamate receptors (*Mays et al., 2009*). Here again, however, in no such case has glutamate been reported to produce muscle activation per se, but rather, this transmitter acts indirectly by regulating ACh's own impact at the NMJ (*Vyas and Bradford, 1987*; *Malomouzh et al., 2003*; *Pinard et al., 2003*) *via* the activation of post-synaptic NMDA receptors (*Pinard and Robitaille, 2008*; *Petrov et al., 2013*). On the other hand, glutamate is the predomi-nant excitatory neurotransmitter in the vertebrate CNS, and supraspinal glutamatergic neurons can re-specify functional glutamatergic NMJs from otherwise purely cholinergic synapses on mammalian skeletal muscles following grafting with transected peripheral motor nerve (*Brunelli et al., 2005*).

Given such a short-term reorganizing capability and the fact that glutamate is a major excitatory neurotransmitter at the NMJ of phylogenetically distant invertebrates (*Gerschenfeld, 1973*), it is possible that the transient employment of glutamatergic neuromuscular transmission in pre-metamorphosing *Xenopus* is representative of a latent ancestral step in the evolutionary transition of intrinsic molecular programming of the NMJ to cholinergic-dependent signaling.

A further and more appealing possibility is that the switching process is related to early appendicular MN axon path-finding and initial NMJ formation because of the unusual context of frog metamorphic development where secondary limb MNs axons must grow to their muscle targets with the primary axial neuromuscular apparatus already in place, fully functional and using ACh as the NMJ transmitter (*Figure 7*). Transplantation experiments in *Xenopus* have demonstrated the ability of MNs to innervate novel tissue targets (*Elliott et al., 2013*). Moreover, amongst the many guidance cues that orient axon elongation during development, both target-derived signals and axon growth cone-released ACh participate in target reaching and initial synapse formation (*Yin and Oppenheim, 1992*; *Erskine and McCaig, 1995*; *Yang and Kunes, 2004*). On this basis, therefore, it is conceivable that the pre-existing axial neuromuscular system in *Xenopus* provides a potential disturbing environment that could attract appendicular motor axons, if they were cholinergic, to make improper neuromuscular connections during pre-metamorphic development. On the contrary, a matched expression of glutamatergic neurotransmitter and receptors in appendicular MNs and limb muscles during pre-metamorphosis could ensure the correct orientation of the developing axon growth cones towards the limb buds and the establishment of synaptic contacts appropriate for the control of limb movements (*Figure 7A*). Supporting this hypothesis are previous findings that morphologically normal neuromuscular connections can still develop despite the experimental suppression of either pre- or postsynaptic cholinergic partners (*Westerfield et al., 1990*; *Misgeld et al., 2002*) and that glutamatergic CNS neurons can replace cholinergic MNs and reinnervate skeletal muscle fibers after a peripheral nerve lesion (*Brunelli et al., 2005*). It is also relevant that in addition to neural signaling, glutamate has been implicated in different neurotrophic functions, including cell growth and migration, during nervous system development (*Nguyen et al., 2001*). In *Xenopus*, once the initial

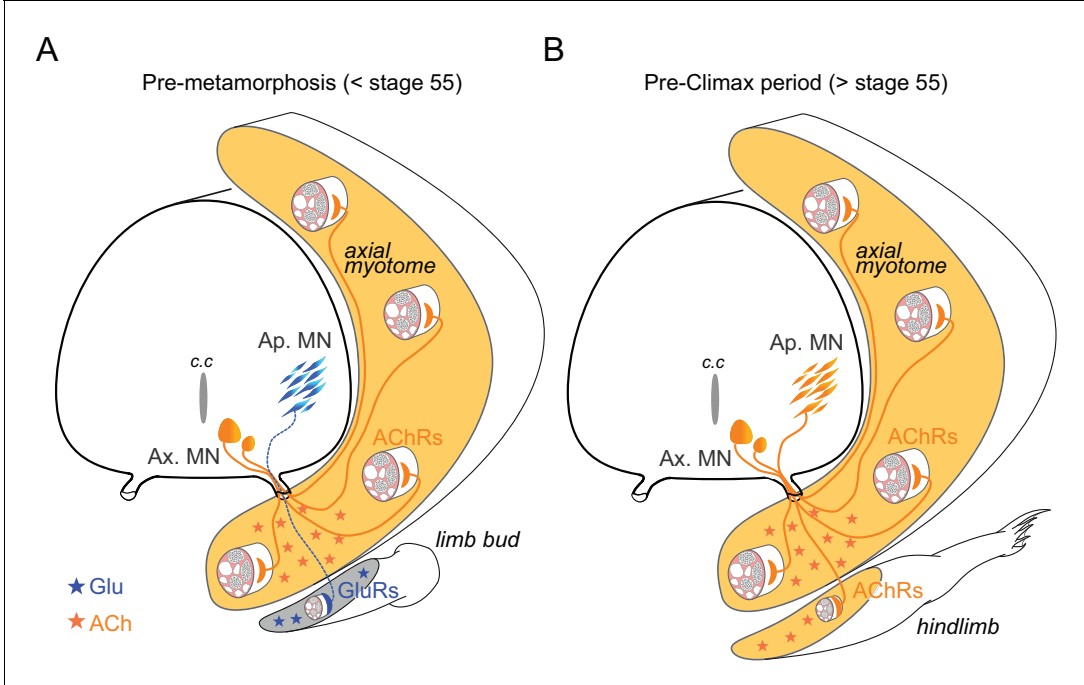

**Figure 7.** Schematic representation of neurotransmitter phenotype switching associated with the establishment of functional limb muscle innervation at different stages of *Xenopus* metamorphic development. See text for further explanation. AChRs: nicotinic ACh receptors; GluRs: glutamate receptors; Ap. MN: appendicular MNs; Ax. MN: axial MNs; c.c: central canal.
DOI: https://doi.org/10.7554/eLife.30693.011

appendicular NMJs are established, both pre- and postsynaptic partners could thereafter be instructed to switch to their definitive cholinergic phenotype (*Figure 7B*).

The signal for such neurotransmitter re-specification remains to be determined, but thyroid hormones, which control the developmental expression of ChAT (*Patel et al., 1987*; *Gould and Butcher, 1989*) and probably also VAChT since both proteins share the same gene locus (*Eiden, 1998*), are most likely to be involved. Consistent with this possibility is that the increase in thyroid hormone levels at metamorphosis onset, which starts at stage 54–55 in *Xenopus* (*Shi, 2000*), triggers a variety of gene-switching molecular programs required for the development of limb muscles and associated motor circuitry through the regulation of spinal cord neurogenesis and functions (*Das et al., 2002*; *Marsh-Armstrong et al., 2004*; *Brown et al., 2005*). Sensory feedback from new proprioceptors in the developing limbs may also participate in the respecification process, as found in the adult rat brain where the occurrence of novel sensory information can trigger neurotransmitter phenotype switching in postsynaptic central neurons (*Dulcis et al., 2013*).

In conclusion, the development of the limb neuromuscular apparatus and limb-based locomotion during *Xenopus* metamorphosis occurs in successive stages that involve a close functional relationship with the preexisting axial motor system until full autonomy is achieved (*Combes et al., 2004*) and, as shown here, is associated with changing underlying molecular patterning. Amongst the latter, neurotransmitter phenotype switching at the NMJ may enable limb motor axons to reach their appropriate muscle targets, which constitutes a novel and fundamental role for this process during motor innervation development (*Spitzer, 2017*) and adds to our understanding of NMJ development in general.

## Materials and methods

### Animals
Experiments were conducted on the South African clawed toad *X. laevis* obtained from the Xenopus Biology Resources Centre in France (University of Rennes 1; http://xenopus.univ-rennes1.fr/). Animals were maintained at 20–22°C in filtered water aquaria with a 12:12 hr light/dark cycle. Developmental stages were sorted according to external body criteria (*Nieuwkoop and Faber, 1956*), and experiments were performed on larvae from stage 49 to 57. All procedures were carried out in accordance with, and approved by, the local ethics committee (protocols #68–019 to HT and #2016011518042273 APAFIS #3612 to DLR).

### Motoneuron retrograde tracing
Procedures used for neuronal retrograde tracing were as described previously (*Bougerol et al., 2015*). Briefly, animals were anesthetized in a 0.05% MS-222 water solution and transferred into a Sylgard-lined Petri dish. In order to backfill MNs from their muscle targets, the skin covering the muscles of interest was dried before a tiny incision was made and crystals of fluorescent dextran amine dyes were applied intramuscularly (*Forehand and Farel, 1982*; *van Mier et al., 1985*; *Roberts et al., 1999*). In most cases, only hindlimb MNs were labeled with either 3 kD rhodamine (RDA) or 10 kD Alexa Fluor 647 (Thermo Fisher, Illkirch, France), except in the experiments illustrated in *Figure 1B,C* where axial MNs were also labeled using 10 kD Alexa Fluor 488. Excess dye was washed out with cold Ringer solution (75 mM NaCl, 25 mM NaHCO$_3$, 2 mM CaCl$_2$, 2 mM KCl, 0.5 mM MgCl$_2$, and 11 mM glucose, pH 7.4). After recovering from anesthesia, larvae were kept in a water tank for 24–48 hr to allow tracer migration into MN cell bodies and dendrites. In a series of experiments (n = 4), the hindlimb buds of stages 51–52 larvae were injected and the animals were kept for several days in a separate aquarium until reaching metamorphic climax (*Figure 2E*), in order to verify that early stage MNs were preserved through later development. Generally, such a labeling approach stains a large proportion of neurons that project processes within the bud muscles and thus potentially labels both MNs and sensory neurons (*Forehand and Farel, 1982*; *van Mier et al., 1985*; *Roberts et al., 1999*). However, since MNs have centrally located cell bodies, our retrograde tracings allowed the confident identification of MN somata only within the spinal cord.

## Immunofluorescence labeling

After MN retrograde labeling, spinal cords were dissected out and fixed in 4% paraformaldehyde (PFA) for 12 hr at 4°C. Preparations were incubated in a 20% [in phosphate-buffered saline (PBS) 0.1%] sucrose solution for 24 hr at 4°C, then embedded in a tissue-tek solution (VWR-Chemicals, Fontenay-sous-Bois, France) and frozen at −45°C in isopentane. 40 μm cross-sections were cut using a cryostat (CM 3050, Leica, Nanterre, France). Fluorescence immunohistochemistry was carried out on these spinal cross-sections using the same protocol as described previously (*Bougerol et al., 2015*). Briefly, after several rinsing steps and the blocking of non-specific sites (using a solution with PBS, Triton X-100 0.3%, bovine serum albumin 1%; Sigma, St. Quentin Fallavier, France) samples were incubated with the primary antibody for 48 hr at room temperature. After rinsing, cross-sections were incubated for 90 min at room temperature with a fluorescently labeled secondary antibody, and washed again before mounting in a homemade medium containing 74.9% glycerol, 25% Coon's solution (0.1M NaCl and 0.01M diethyl-barbiturate sodium in PBS), and 0.1% paraphenylene-diamine. The primary antibodies used were goat anti-ChAT (1:100; Millipore), rabbit anti-VAChT (1:1000; Santa-Cruz) and mouse anti-*Islet*1/2 (1:250; Developmental Studies Hybridoma Banks (DHSB), University of Iowa, Iowa city, US). Secondary antibodies were donkey anti-goat and anti-rabbit or anti-goat and anti-mouse IgGs coupled to Alexa Fluor 488 and 568 (1:500; Life Technologies).

Fluorescent immunohistochemistry was carried out on entire or 20 μm sliced hindlimb buds from developmental stages 51 to 57 after overnight fixation in PFA 4%. The same labeling protocol was used as described above for spinal cross-sections. Alexa Fluor 488-conjugated α-bungarotoxin (10 μg/ml; Life Technologies) was used to label neuromuscular junctions. The primary antibody, mouse anti-neurofilament associated protein (3A10; 1:100; DHSB), was used to label nerve branches innervating the limb bud. The primary antibodies, rabbit anti-synaptophysin (1:500; abcam, Paris, France), goat anti-NMDA subunit one receptor (NR1; 1:100; abcam) or rabbit anti-NR2b (1:200; abcam), and guinea pig or mouse anti-VGluT1 (1:100; respectively from Millipore, France and Synaptic Systems, Germany), were used to label synapses, glutamate receptors, and vesicular glutamate transporters, respectively. Note that comparable results were obtained with both anti-VGluT1 antibodies. The bud preparations were then incubated with secondary antibodies donkey anti-mouse, anti-rabbit and anti-goat IgGs coupled to Alexa Fluor 488, 568 and 647 (1:500, Thermo Fisher). For microscope imaging whole-mount limb buds were mounted on cavity slides in a homemade medium (see above).

## Image acquisition and fluorescence quantification

Whole-mount preparations and cross-sections labeled with fluorescent material were imaged using an Olympus FV1000 confocal microscope equipped with 488, 543 and 633 nm laser lines. Images were processed using Fiji and Photoshop softwares. Multi-image confocal stacks with 1 μm z-step intervals were generated using a 20x/0.75 oil objective and with 0.3 μm z-step intervals using a 60x/1.4 oil objective. Figure images were obtained by orthogonal projection from multi-image stacks with artificial fluorescent colors using the freeware Fiji.

ChAT, VAChT and *Islet*1/2 fluorescence quantifications were performed on original images from 0.3 μm z-step stacks. Fluorescence intensity was measured automatically from 3 ROI in single planes with the same size, defining the slice background, and the axial and appendicular MNs fluorescence signals, respectively. The variation of fluorescence ($\Delta F/F = (F-F_0)/F_0$) was calculated for both axial and appendicular MNs ROI relative to background, the latter being acquired in a ventral spinal region devoid of axial and limb MN cell bodies, yet where cholinergic terminals were present. This calculation was performed on five consecutive confocal planes per slice where the appendicular motor column was identified by retrograde labeling (usually 2 to 5 40 μm slices) and where background noise was maximal. Preparations with concomitant ChAT and VAChT fluorescent labeling were combined for statistical quantification (*Figure 2C*).

## In situ hybridization

Because of its tetraploid status, *X. laevis* possesses two genes for ChAT (Genbank accession numbers: XM_018225547 and XM_018239425 for ChAT1 and ChAT2, respectively) that exhibit about 87% nucleotide identity. This prompted us to generate two ChAT probes. To synthesize these probes, two PCR fragments of 529 (ChAT1) and 512 (ChAT2) bp were amplified from adult *X. laevis*

brain and spinal cord RACE-ready cDNA (*Bougerol et al., 2015*), then subcloned into pGEM-T easy (Promega, Charbonnières, France). The pairs of primers used were: 5′-TTTGCTGCCAACCTTATCTC TG-3′ and 5′-ATGAAGTAACTGCAAAGCCCTG-3′ for ChAT1, and 5′-TCTGGAGTGCTGGATTACAA-3′ and 5′-ATGAAGTAACTGCAAAGCCCTG-3′, for ChAT2. Sense and antisense digoxigenin-labeled riboprobes were synthesized from the linearized plasmid with the RNA polymerases T7 or Sp6, using the RNA Labeling Kit (Roche Diagnostics, Mannheim, Germany).

The trunk region of tadpoles from stages 49 to 55 was dissected, fixed with 4% PFA in 0.1 M PBS overnight at 4°C and rinsed in 0.1 M PBS. Fixed samples were cryoprotected in 15% then 30% sucrose/PBS and embedded in Tissue-Tek (Sakura, Netherlands). Frontal sections (20 µM) of the trunks were cut at −20°C using a cryostat, collected on Superfrost Plus slides (O. Kindler, Freiburg, Germany), dried at room temperature for 24 hr and stored at −80°C until use.

The in situ hybridization protocol used in the present study was adapted from earlier studies (*Buresi et al., 2012*; *Bougerol et al., 2015*) and consisted of the following steps. Briefly, sections were rinsed 2 × 5 min with PBS at room temperature, 15 min in five times concentrated sodium chloride and sodium citrate solution (5X SSC), then were incubated for 2 hr in prehybridization buffer (50% formamide, 5X SSC, 50 µg/ml heparin, 5 mg/ml yeast RNA, 0.1% Tween) at 65°C. When prehybridization was complete, the prehybridization solution was removed and replaced with the same buffer containing a mix of the two heat-denatured digoxigenin-labeled ChAT1 and ChAT2 riboprobes. Hybridization was carried out overnight at 65°C. Sections were rinsed 3 × 30 min in 2X SSC at 65°C then 1 hr in 0.1 X SSC at 65°C. Two final washes were performed for 5 min in MABT (maleic acid 100 mM, pH 7.2, NaCl 150 mM, Tween 0.1%) at room temperature. Sections were transferred to a blocking solution [5% blocking reagent (Roche), 5% normal goat serum in MABT] and incubated at room temperature for 1 hr, before addition of the alkaline phosphatase coupled anti-digoxigenin antibody (1/4000) for overnight storage at 4°C. Sections were again washed 3 × 10 min in MABT and 2 × 5 min in PBS at room temperature, then incubated 10 min in staining buffer (100 mM Tris-HCl, pH 9.5, 50 mM MgCl$_2$, 100 mM NaCl, and 0.1% Tween) and transferred to BM Purple (Roche) for colorimetric detection. Finally, they were washed twice in PBS to stop the reaction, and then mounted on gelatin-coated slides in Mowiol. Sections were imaged using a Leica DM5500 B microscope connected to LAS V4.1 software. The specificity of the hybridization procedure was verified by incubating sections with the sense riboprobes with which only background signals typical of this type of chemical reaction could be observed.

## Patch-clamp recording of appendicular motoneurons

Retrograde labeling of appendicular MNs was performed using 3kD RDA dextran on stages 51, 52 larvae. The day after, patch-clamp electrophysiological recordings of labeled MNs were made on isolated brainstem-spinal cord in vitro preparations (n = 7). After anesthesia in 0.05% MS-222, the brainstem and spinal cord, including spinal segmental ventral roots, were dissected out in cold oxygenated (95% O$_2$, 5% CO$_2$) Ringer solution. The preparation was then placed in a recording chamber and continuously superfused with oxygenated Ringer solution (~2.5 mL, ~17°C, rate of ~2 mL/min). Spontaneous fictive locomotor episodes were recorded from a caudal ventral root (between segments 12 and 15; *Figure 3B*) using a borosilicate glass suction electrode (tip diameter, 100 nm; Clark GC 150F; Harvard Apparatus) filled with Ringer solution. The recorded signal was amplified (A-M system), rectified and integrated (time constant 100 ms; Neurolog System). RDA-positive appendicular MNs were identified with a standard epifluorescent illumination system (Cy3 filter) within the whole-mount spinal cord (dorsal-side opened) and subsequently visualized using a differential interference contrast microscope with an infrared video camera to facilitate the patch electrode trajectory (*Figure 3B*). Using an Axoclamp 2A amplifier (Molecular Devices, Berkshire, UK), whole-cell patch-clamp recordings were made with a borosilicate glass electrode (pipette resistance, 5–6 MΩ; Clark GC 150TF; Harvard Apparatus) filled with a solution containing (in mM) 100 K-gluconate, 10 EGTA, 2 MgCl$_2$, 3 Na$_2$ATP, 0.5 NaGTP, 10 HEPES, pH 7.3. In additional experiments (n = 2), to impose an elevated intracellular chloride concentration corresponding to that of immature neurons (*Ben-Ari, 2002*), including those in *Xenopus* (*Akerman and Cline, 2006*), this low [Cl$^-$] recording solution was replaced by a high [Cl$^-$] version that contained (in mM) 70 K-gluconate, 30 KCl, 10 EGTA, 2 MgCl$_2$, 3 Na$_2$ATP, 0.5 NaGTP, 10 HEPES, pH 7.3. Alexa Fluor 488 (Life Technologies), which was added in the patch pipette to fill the recorded neurons, allowed subsequent verification that recorded cells were indeed RDA-positive (*Figure 3B*). All electrophysiological signals

were computer-stored using a digitizer interface (Digidata 1440; Pclamp10 software; Molecular Devices) and analyzed offline with Clampfit software (Molecular Devices).

## Axial and hindlimb EMG recordings in semi-intact preparations

Semi-intact preparations from stages 52 to 57 larvae were used to simultaneously record EMG activity from axial myotomes and hindlimb bud muscles (*Figure 4*; n = 12). Brainstem-spinal cord preparations were dissected out in the same way as for patch-clamp recording, but tail myotomes (7-10) and hindlimb buds were left attached to the spinal cord. Semi-intact preparations were fixed in a Sylgard-lined recording chamber, continuously superfused with oxygenated Ringer solution (1.3–2.1 mL/min) and maintained at $18 \pm 0.1°C$ with a Peltier cooling system. In some experiments, *d*-tubocurarine (30 µM; Sigma) was exogenously applied to block nicotinic receptor-type cholinergic synapses. Fictive locomotion sequences were generated either spontaneously or triggered by electrical stimulation of the caudal region of the brainstem (Grass stimulator S88), and the spinal swimming pattern was monitored from a caudal ventral root (between segments 12 and 15) with a suction electrode as described above. EMG activities in rostral myotomes (7–10) and hindlimb buds were recorded simultaneously using pairs of 50 µm insulated wire electrodes connected to a differential AC amplifier (A-M System). Both nerve and EMG activities were digitized at 10 kHz (CED 1401, Cambridge Electronic Design, UK), and displayed and stored on computer for offline analysis with Spike2 software (CED).

Discharge rates in individual nerve and EMG recordings were measured by setting an amplitude threshold to count all impulses in such multi-unit recordings. Firing rates (in spikes/s) were averaged over 10–20 locomotor cycles. Cycle period was taken as the interval between the onsets of two consecutive ventral root bursts. These consecutive burst onsets were used as a trigger for averaging the discharge rates of each EMG channel over cycle duration.

## EMG recordings and drug applications in isolated nerve-limb bud preparations

Isolated hindlimb-bud preparations with the sciatic and crural nerves still attached (*Figure 6*; n = 17) were used to record the EMG activity evoked by electrical stimulation of either appendicular nerve branch at stages 52 to 57. Under MS-222 anesthesia, appendicular nerves were disconnected from the spinal cord and separated from tail myotomes, taking care not to detach them from the rest of the bud. The bud and attached nerves were fixed with small pins in a Sylgard-lined recording chamber and superfused with oxygenated Ringer solution. A small incision was made at the distal extremity of the bud to allowing insertion of the EMG electrodes. Either limb nerve branch was stimulated with a glass suction electrode connected to a Grass stimulator S88 through a photoelectric stimulus isolation unit (PSIU 6; Grass Instruments). Note that stimulating either branch provided similar results, and no distinction was made in this report. Single pulses (70–300 µA; 10 µs) were delivered every 100 s, and polarity inversion tests were performed in order to distinguish the stimulation artifact from the muscle response. EMG signals were amplified, integrated (time constant 10 ms) with Spike2 software and stored as described above. EMG responses were measured as the area under the integrated recording traces.

30 µM *d*-tubocurarine or a cocktail of 20 µM CNQX +10 µM AP5 were added to the perfusion solution to block nicotinic or glutamatergic receptors, respectively (all drugs from Sigma). Because of the typical difficulty in washing-out these drugs in such experiments, a second antagonist was generally added while the first one was still present in the bath. Control experiments where only one drug was applied showed effects similar to those of combined drug application, and thus data from either approach were pooled in this study. In some experiments on early stages, the normal Ringer solution was replaced by a solution containing half (1 mM) of the normal concentration of $MgCl_2$ in order to unmask any NMDA receptor-mediated component of the EMG's glutamatergic response. Despite causing a noticeable increase in the control EMG amplitude, such a low $Mg^{2+}$ solution had no influence on the effects resulting from subsequent antagonist application.

## Statistics

After signal processing in Spike2, electrophysiological data were analyzed using Prism5 (GraphPad, USA) and OriginPro8 (OriginLab Corporation, USA). Data are shown as means and standard errors

of the mean (± SEM), unless stated otherwise. For ChAT and VAChT immunofluorescence signals (*Figure 2C*), differences between preparation means were tested using the Kruskall-Wallis ANOVA multi-comparison test (Student-Newman-Keuls method; statistical significance is indicated in *Figure 2C*). For integrated EMG signals, differences of means were tested using the unpaired two-tailed Mann–Whitney U-test (statistical significance and U values are indicated in the corresponding Result section).

## Acknowledgements

We are grateful to Gilles Courtand (UMR 5287) and Cyril Willing (Centre de Microscopie de fluorescence et d'Imagerie numérique du Muséum) for technical assistance in image analysis, Feng Quan, Anne-Laure Gaillard (both UMR 7221) and Boudjema Imarazene (UMR 7208) for technical assistance with in situ hybridization, and Lionel Parra-Iglesias (UMR 5287) for animal care. This work was supported by grants from the *Centre National de la Recherche Scientifique*, by the *Actions Thématiques du Muséum* and the Fondation pour la Recherche Médicale (Equipe FRM DEQ20170336764 M. Thoby-Brisson).

## Additional information

### Funding

| Funder | Grant reference number | Author |
|---|---|---|
| Fondation pour la Recherche Médicale | Equipe FRM DEQ20170336764 | Muriel Thoby-Brisson |
| Muséum National d'Histoire Naturelle | Actions thématiques du Museum | Hervé Tostivint |
| Centre National de la Recherche Scientifique | | Didier Le Ray |

The funders had no role in study design, data collection and interpretation, or the decision to submit the work for publication.

### Author contributions

Francois M Lambert, Didier Le Ray, Conceptualization, Resources, Data curation, Software, Formal analysis, Supervision, Validation, Investigation, Visualization, Methodology, Writing—original draft, Project administration, Writing—review and editing; Laura Cardoit, Elric Courty, Data curation, Formal analysis, Methodology; Marion Bougerol, Data curation; Muriel Thoby-Brisson, Conceptualization, Data curation, Formal analysis, Methodology, Writing—original draft, Writing—review and editing; John Simmers, Writing—original draft, Writing—review and editing; Hervé Tostivint, Conceptualization, Data curation, Formal analysis, Supervision, Methodology, Writing—original draft, Writing—review and editing

### Author ORCIDs

Francois M Lambert http://orcid.org/0000-0002-8655-2652
Muriel Thoby-Brisson http://orcid.org/0000-0003-3214-1724
Didier Le Ray http://orcid.org/0000-0003-2089-9861

### Ethics

Animal experimentation: All procedures were carried out in accordance with, and approved by, the local ethics committee (protocols no. 68-019) to H. Tostivint and no. 2016011518042273 APAFIS no. 3612 to DLR)

### Decision letter and Author response

Decision letter https://doi.org/10.7554/eLife.30693.016
Author response https://doi.org/10.7554/eLife.30693.017

# Additional files

## Supplementary files

• Transparent reporting form
DOI: https://doi.org/10.7554/eLife.30693.012

## Data availability

All data generated or analysed during this study are available via Dryad (doi:10.5061/dryad.9sj250q).

The following dataset was generated:

| Author(s) | Year | Dataset title | Dataset URL | Database, license, and accessibility information |
|---|---|---|---|---|
| Lambert FM, Cardoit L, Courty E, Bougerol M, Thoby-Brisson M, Simmers J, Tostivint H, Le Ray D | 2018 | Data from: Functional limb muscle innervation prior to cholinergic transmitter specification during early metamorphosis in Xenopus | https://dx.doi.org/10.5061/dryad.9sj250q | Available at Dryad Digital Repository under a CC0 Public Domain Dedication |

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
