## [Decision Letter]

Thank you for submitting your article "Functional limb muscle innervation prior to cholinergic transmitter specification during early metamorphosis in *Xenopus*" for consideration by *eLife*. Your article has been reviewed by three peer reviewers, and the evaluation has been overseen by a Reviewing Editor and Eve Marder as the Senior Editor.

The reviewers have discussed the reviews with one another and the Reviewing Editor has drafted this decision to help you prepare a revised submission.

The reviewers found your study on changes in limb motor neuron transmitter phenotype and the composition of receptors at the frog neuromuscular junction of interest and well-executed. The reviewers welcomed your evidence that during metamorphosis when the limbs are forming and becoming innervated, motoneurons initially release glutamate, and then acetylcholine at their synapses on limb musculature. They agree that the manuscript is clearly written and for the most part well-illustrated, especially the use of retrograde labeling to identify and distinguish appendicular and axial motoneurons.

However, the three reviewers concurred on amending four aspects, all of which would greatly strengthen your study and begin to shed light on mechanism. They expected that these amendments could be accomplished within *eLife*'s expected timeline for revisions:

1) Demonstration of the presence of any of the known vesicular glutamate transporters in motoneurons at the expected ages.

2) Better explanation in the text of the large effect of d-tubocurarine. There is a discrepancy in the reported effects of applying t-tubocurarine at Stage 53, leading to a large decrease in amplitude of EMG activity, and the localization of receptors. Reviewer 2 asks whether you could test the effects of tubocurarine or glutamate antagonists at an earlier stage (e.g., 52).

3) Inclusion in the Results section of the electrophysiological recordings showing the effects of glutamatergic antagonists. This experiment was alluded to in the Materials and methods section in which a cocktail of CNQX and AP5 was used to block glutamatergic receptors and are critical to the arguments of the paper.

4) Better and more complete statistical analyses: Reviewer 3 notes a lack of statistical analysis and an indication of the number of experiments used to generate the quantitative data, a lack of statistical comparison between the fluorescence levels at the different ages, and confusion over standard error of the mean and standard deviation used to illustrate the variability.

*Reviewer #1:*

1) Subsection “Locomotor-related activation of hindlimb muscles by non-cholinergic motoneurons”, "is initially largely independent of ACh signaling in…" The amplitude is reduced by more than 50% by d-tubocurarine. How about "is initially only partially suppressed in…."? Could the residual signals be stimulus artifacts? In contrast to Control and Wash, the amplitudes are constant in amplitude. The Materials and methods section describes experiments in which a cocktail of CNQX and AP5 was used to block glutamatergic receptors. These experiments should be illustrated here. They are critical to the story.

Related to this point, subsection “Developmental switch in hindlimb neuromuscular transmission”, "indicating the absence of anatomically detectable muscle ACh receptors" would be more appropriate. Figure 4B suggests that they are detectable by physiological / pharmacological methods.

Again, related to this point, in the Discussion section, I find "indicative of a complete neurotransmitter phenotype switching…" too great a claim, given that d-tubocurarine substantially reduces the amplitude of the l-HLemg in Figure 4B.

2) Discussion section, the authors argue against a muscle spindle innervation role for the limb motoneurons they have studied, on the basis of the sizes of the muscle spindle- and muscle fiber-innervating populations of neurons. However, in the absence of information about the sizes of these populations this is not convincing.

3) Quantification is inadequate. The numbers of animals in which observations were made should be provided in each figure legend. Contrary to the assertion in the Transparent Reporting Form, the number of replicates is largely omitted in the Results section and Figure Legends.

*Reviewer #2:*

1) Figure 4: It is argued that the switch from glutamatergic to cholinergic transmission occurs around stages 54-55. However, the results presented in Figure 4 clearly show that cholinergic transmission is at place already at stage 53 and possibly even at earlier stages. There is a large decrease in the amplitude of the EMG activity by d-tubocurarine.

Could these experiments be done at stage 52? Alternatively, is it experimentally possible to apply glutamatergic antagonists locally on the limb bud?

The results of Figure 4 show clearly that the EMG activity at stage 53 is largely mediated by t-tubocurarine-sensitive receptors. This is at odds with the main conclusion of this study.

2) Figure 5C,D: The EMG activity recorded at stage 53 is very weak and this may indicate that the neuromuscular junction is not fully developed.

The authors should provide electrophysiological traces showing the effects of glutamatergic antagonists. Also, the weak EMG activity recorded at stage 53 might be due to the absence of activation of NMDA receptors. I suggest to repeat these experiments using a Mg-free extracellular solution.

3) Overall, it is not clear if limb motor neurons switch the transmitter phenotype or if they use both glutamate and acetylcholine as co-transmitters with a dominance of the one or the other at different developmental stages. In this regard, the authors could test if limb motor neurons co-express transporters for glutamate and acetylcholine in different proportions at different stages.

*Reviewer #3:*

The central thesis of this paper is that motoneurons transiently release glutamate instead of acetylcholine at the neuromuscular junction of developing hindlimb muscles at the onset of metamorphosis. The main evidence in support of this conclusion comes from the immunocytochemistry showing that appendicular motoneurons do not express either ChAT or VAChT immunoreactivity and exhibit very little ChAT mRNA at stage 53. Moreover, at stage 52, the developing limb muscle has very low levels of α-bungarotoxin but does express NMDA receptor subunits some of which are co-localized with neuronal synaptophysin (although many are not).

Unfortunately, the physiological experiments do not completely support the anatomy. For example, at stage 53 bath application of d-tubocurarine greatly reduces the amplitude of the rhythmic EMG signals from the limb bud during locomotor episodes despite the apparent absence of acetylcholine receptors in the muscle and cholinergic markers in limb motoneurons. The authors do not explain why the EMG is reduced at this age. Similarly, in the nerve/limb preparation illustrated in Figure 5, bath application of CNQX/AP5 only reduces the stimulus-evoked EMG by 50%. Why is the block incomplete? Again, this is not discussed by the authors. The authors showed only the presence of NMDA receptor subunits in the muscle but did not show the effect of NMDA antagonists alone on the evoked EMG to validate their presence.

In both experiments, it would have been useful to combine the glutamatergic and cholinergic antagonists to see if this produced a complete blockade of the muscle electrical activity.

Finally, in Figure 5D, the results for control and d-tubocurarine are shown, but it would also be of relevance to show a recording for the glutamate antagonists. Here, the number of experiments is missing, and no p-values are reported.

The intracellular recordings of the locomotor activity illustrated in Figure 3E are puzzling. The recordings reveal rhythmic hyperpolarizing potentials and the cells do not fire. Was this true of all of the motoneurons that were recorded? If so, how are the muscles activated if the motoneurons do not spike? For the 11 recorded motoneurons, the authors should indicate from how many preparations they were obtained, and if the motoneurons were only identified by their labeling or whether they tested for an antidromic response. If it is the former, did the authors ever use a second dye applied intracellularly to confirm that the labelled motoneuron was indeed the one that was recorded from.

At the earliest stages of limb formation, it was stated that motoneurons were small cells. However, the resistance of the cell shown in Figure 3C was between 3 to 5 MOhm suggesting either a very leaky cell or a large one. What was the input impedance of the other motoneurons?

As already alluded to, a major concern is the lack of statistical analysis and an indication of the number of experiments used to generate the quantitative data. For example, in subsection “Delayed cholinergic transmitter phenotype expression in the developing appendicular motor Column” there is no statistical comparison between the fluorescence levels at the different ages nor is the number of experiments shown. Similar problems apply to the quantification in other parts of the paper. The authors often refer to the mean and the SE. Do they mean the standard error of the mean (S.E.M). Why was the standard error of the mean rather than the standard deviation used to illustrate the variability?

The quantification of the fluorescence in Figure 2C was confusing. In the text, it states that ChAT and VAChT were not immuno-detected in appendicular MNs until stage 55. The figures in Figure 2A confirm this. However, in the plot of 2C (52-54), which shows the AP/AX fluorescence ratio, the values range from ~0-14% for ChAT. It is difficult to see how these values are generated if AP is zero. The authors should indicate how the ratio was calculated.

Also, despite an increase in the ratio at later stages, it is still about 20% of the axial fluorescence (when appendicular MNs are supposed to be bigger). This implies that the majority of appendicular motoneurons are still not cholinergic. It would be helpful to show that once the metamorphosis is complete, this ratio is 100% thereby validating the measure.

I found the in-situ hybridization results shown in Figure 2D, difficult to interpret. It is difficult to know the extent of the ChAT ISH labelling in these Figures For example, the extent of the Axial motor column in D 53 looks much more limited than the ChAT labelling in A 53. Furthermore, it is not clear that appendicular MNs at stage 53 are negative. The background is very high. Sections with dextran labelling would have made clear where the motoneurons are located. This figure does not agree with the statement that ISH is more sensitive than immunodetection. How many times was this experiment performed?

[Editors' note: further revisions were requested prior to acceptance, as described below.]

Thank you for resubmitting your work entitled "Functional limb muscle innervation prior to cholinergic transmitter specification during early metamorphosis in *Xenopus*" for further consideration at *eLife*. Your revised article has been favorably evaluated by Eve Marder (Senior Editor), a Reviewing editor, and three reviewers.

The manuscript has been improved and reviewers 1 and 2 were satisfied with your amendments, but there are some remaining issues raised by reviewer 3 that remain to be addressed, indicated in the appended reviews. The other reviewers and I in consultation now agree on the need for these revisions:

1) The verity of the VGluT1 localization, in synaptic terminals rather than in cell bodies; the predicted restriction of vGluT1 to motor neuron terminals and colocalization with synaptic proteins, and mRNA localization in motor neuron cell bodies; and a comparison to IHC of vGluT2. These aspects were deemed essential to your next revision.

The next two sets of comments could be dealt with in the text of the Discussion section:

2) incomplete blockade of evoked muscle action by glutamate antagonists, that could be explained by the low concentration of glutamate antagonists used;

3) physiology experiments showing that the reduction of the calcium transients generated during fictive swimming by tubocurarine may be due to blockade of GABA-A or glycinergic receptors rather than cholinergic receptors.

*Reviewer #1:*

The authors have addressed my concerns satisfactorily. Demonstration of the presence of vGluT1 and the effective use of glutamate receptor antagonists were key. The data in Figure 6B,C are nicely characterized as a "functional switch in hindlimb neuromuscular transmission".

*Reviewer #2:*

The authors revised this manuscript according to the criticisms raised by the reviewers. The responses provided reasonable explanations and interpretations.

*Reviewer #3:*

In response to the reviewers' concerns the authors have attempted to demonstrate that motoneurons express VGluT1. Unfortunately, I did not find their presentation of the data convincing. For example, VGluT1 expression at stage 52 (Figure 5—figure supplement 1) is widespread throughout the ventral cord and appears, surprisingly, to label cell bodies. This is in contrast to numerous other immunocytochemical studies of VGluT1 labelling in the spinal cord that show the labelling is restricted to synaptic terminals. (Betley et al., 2009; Schultz et al., 2017). The authors should illustrate an image of the whole transverse section of the cord to show how widespread the labelling is. Was this antibody tested or has it been described in other papers to show that it is indeed targeting VGluT1? What experiments did the authors perform to examine the specificity of the antibody? The VGluT1 labelling shown in Figure 5B appears coextensive with the muscle nerve but there is no evidence of the more punctate labelling that characterizes the neuromuscular junction and is shown in Figure 5A (α-bungarotoxin). Furthermore, the authors should show the VGluT1 labelling at stage 57 when glutamatergic transmission is absent. Finally, it was not clear if the authors had tried to label the cells with VGluT2 whose mRNA has been reported in some studies to be expressed in the neonatal mammalian motoneurons.

A more convincing approach to demonstrate the presence of a vesicular protein in motoneurons would be to show that it co-localizes with a synaptic protein – such as synaptophysin at the neuromuscular junction or to demonstrate the presence of the mRNA in motoneuronal cell bodies.

---

## [Author Response]

Essential revisions:

1) Demonstration of the presence of any of the known vesicular glutamate transporters in motoneurons at the expected ages.

New experiments on the immuno-detection of VGluT1 have demonstrated the presence of the glutamate vesicular transporter in early stage tadpoles, both at the level of motor axons in the developing hindlimbs (illustrated in new Figure 5C) and the limb motoneuron cell bodies themselves (illustrated in new Figure 5—Figure supplement 1).

2) Better explanation in the text of the large effect of d-tubocurarine. There is a discrepancy in the reported effects of applying t-tubocurarine at Stage 53, leading to a large decrease in amplitude of EMG activity, and the localization of receptors. Reviewer 2 asks whether you could test the effects of tubocurarine or glutamate antagonists at an earlier stage (e.g., 52).

Evidently there was a general misinterpretation of the results illustrated in Figure 4 and original Figure 5, and we thank the three referees for having brought this lack of clarity to our attention. In old Figure 5 (new Figure 6), we used isolated hindlimb preparations to stimulate directly the limb motor nerve, and therefore the antagonists’ effects on the EMG responses could be concluded to be exclusively peripheral, at the neuromuscular junction. In Figure 4, in contrast, the experiments were performed on semi-intact preparations where spontaneously-active spinal CPG circuitry was still present and able to drive motoneurons and EMG activity in both the axial and limb muscles. In this case, any observed effects of *d*-tubocurarine could be attributable to an action at either central or peripheral levels, or both. In comparing the results of the experiments illustrated in Figure 4 and old Figure 5, it seemed reasonable to us to interpret the EMG amplitude changes seen under *d*-tubocurarine at stage 53 as being centrally-mediated in Figure 4 and peripherally-mediated in old Figure 5. Manifestly, however, this was not obvious to the reader. We now also provide new data (in Figure 4—figure supplement 1) showing that cholinergic synapses are present on limb motoneuron cell bodies, that they play an important role in the excitation of these motoneurons during centrally-generated swimming and are directly affected by nicotinic antagonist perfusion. This is demonstrated by new calcium imaging experiments that show a strong reduction in the calcium signal in limb motoneurons during fictive swimming in the presence of d-tubocurarine. These supplementary results, which are briefly described in subsection “Locomotor-related activation of hindlimb muscles by non-cholinergic motoneurons”, thus strongly support the conclusion that d-tubocurarine was indeed having a major central effect on limb motoneuron activation in the experiments of Figure 4B (as corroborated in old Figure 5/new Figure 6 by the responses that could only occur at the level of the neuromuscular junction).

3) Inclusion in the Results section of the electrophysiological recordings showing the effects of glutamatergic antagonists. This experiment was alluded to in the Materials and methods section in which a cocktail of CNQX and AP5 was used to block glutamatergic receptors and are critical to the arguments of the paper.

These data are now illustrated in new Figure 6. We initially believed that illustrating the cholinergic drug effects was important for emphasizing more the non-cholinergic nature of the transmitter phenotype present at early developmental stages. However, we agree that illustrating the parallel effects of glutamate antagonists does strengthen the phenotype switching message of the article.

4) Better and more complete statistical analyses: Reviewer 3 notes a lack of statistical analysis and an indication of the number of experiments used to generate the quantitative data, a lack of statistical comparison between the fluorescence levels at the different ages, and confusion over standard error of the mean and standard deviation used to illustrate the variability.

The lack of numbers of experiments in the original manuscript was due to an unfortunate editing error for which we apologize. This information is now reestablished, where appropriate, in the text and/or figure legends. The lack of statistical comparison pointed out by reviewer 3 relates specifically to the changes in ChAT and VAChT fluorescence levels found in limb motoneurons during development (Figure 2C). Although we originally performed such a comparison, we didn't report it because of uncertainty about its validity due to prepto-prep differences, for example, in the extent of retrograde label uptake, fluorescence intensities, and tissue penetration by both primary and secondary antibodies. However, as requested by the reviewer, this comparative analysis is now provided in Figure 2.

We have followed conventional procedure in providing means ± SEM in the text. In original Figure 2C, individual values (circled dots) were presented for single experiments in each age group, allowing their dispersion to be visualized, with the error bars representing the standard error of the overall mean (black dots) of these values per group. Note that this analysis has been redone according to reviewer 3’s remarks and that we have changed the data display to improve clarity.

Reviewer #1:

1) Subsection “Locomotor-related activation of hindlimb muscles by non-cholinergic motoneurons”, "is initially largely independent of ACh signaling in…" The amplitude is reduced by more than 50% by d-tubocurarine. How about "is initially only partially suppressed in…."? Could the residual signals be stimulus artifacts? In contrast to Control and Wash, the amplitudes are constant in amplitude. The Materials and methods section describes experiments in which a cocktail of CNQX and AP5 was used to block glutamatergic receptors. These experiments should be illustrated here. They are critical to the story.

This part of the Results section has been rewritten to incorporate new immunohistochemical and electrophysiological data (Figure 4—figure supplement 1) which indicate that the reduction in limb muscle EMG under d-tubocurarine in pre-stage 55 preparations was most likely due to a central nervous effect on motoneuron activation rather than a peripheral effect on the neuromuscular junction. Note also that the activity recorded in Figure 4 is spontaneous fictive swimming without involving any electrical stimulation (and thus the possibility of stimulus artifacts).

Hindlimb EMG responses to motor nerve electrical stimulation in the presence of glutamatergic antagonists are now illustrated in new Figure 6 for both early and late developmental stages, as requested by the three reviewers and the corresponding text have been modified (subsection “Developmental switch in hindlimb neuromuscular transmission").

Related to this point, subsection “Developmental switch in hindlimb neuromuscular transmission”, "indicating the absence of anatomically detectable muscle ACh receptors" would be more appropriate. Figure 4B suggests that they are detectable by physiological / pharmacological methods.Again, related to this point, in the Discussion section, I find "indicative of a complete neurotransmitter phenotype switching…" too great a claim, given that d-tubocurarine substantially reduces the amplitude of the l-HLemg in Figure 4B.

In order to be more circumspect, we have rewritten as “indicating an absence of muscle ACh receptors…” subsection “Developmental switch in hindlimb neuromuscular transmission”.

For the same reason, “indicative” has been replaced by “consistent with” in the Discussion section.

We agree that Figure 4B could suggest ACh to be involved in the neuromuscular transmission at stage 53. However, as stated in our response to the reviewer’s preceding comment, another explanation is that the observed effect resulted from the d-tubocurarine acting centrally on limb motoneurons during fictive axial swimming (see new Figure 4—figure supplement 1 and the new paragraph in subsection “Locomotor-related activation of hindlimb muscles by non-cholinergic motoneurons). Specifically, these new data show the presence of cholinergic synapses in the vicinity of limb MN cell bodies and a suppressive effect of the nicotinic antagonist on the calcium dynamics of limb MNs during fictive swimming. Taken together with our other results, therefore, these new data add strong support to the conclusion of a complete switch in the neurotransmitter phenotype of limb motoneurons during early metamorphosis.

2) Discussion section, the authors argue against a muscle spindle innervation role for the limb motoneurons they have studied, on the basis of the sizes of the muscle spindle- and muscle fiber-innervating populations of neurons. However, in the absence of information about the sizes of these populations this is not convincing.

A specific fusimotor innervation has not previously been described in frogs, other than by Fujitsuka et al., (1987) who reported the existence of a small population of fusimotor MNs innervating exclusively muscle spindles, and in only one muscle (the semitendinosus). Although not quantified, our data on Islet immunoreactivity (and now from *in situ* hybridization; Figure 3—figure supplement 1) have revealed the existence of a relatively large population of putative appendicular MNs, at least the greater proportion of which was identified (by back-filling) as projecting into the limb bud. Thus, it seems reasonable to conclude that this extensive population of non-cholinergic limb MNs does not correspond to a possible discrete fusimotor population. The text has been rewritten (Discussion section) to try and clarify this point.

3) Quantification is inadequate. The numbers of animals in which observations were made should be provided in each figure legend. Contrary to the assertion in the Transparent Reporting Form, the number of replicates is largely omitted in the Results section and Figure Legends.

We apologize for this omission, which has been corrected. All numbers of animals and cells are now provided where appropriate in the text and/or figure legends.

Reviewer #2:

1) Figure 4: It is argued that the switch from glutamatergic to cholinergic transmission occurs around stages 54-55. However, the results presented in Figure 4 clearly show that cholinergic transmission is at place already at stage 53 and possibly even at earlier stages. There is a large decrease in the amplitude of the EMG activity by d-tubocurarine.Could these experiments be done at stage 52? Alternatively, is it experimentally possible to apply glutamatergic antagonists locally on the limb bud?The results of Figure 4 show clearly that the EMG activity at stage 53 is largely mediated by t-tubocurarine-sensitive receptors. This is at odds with the main conclusion of this study.

This important point was also raised by reviewer 1. Our response is the same and is as follows: We agree that Figure 4B could suggest that ACh is involved in transmission at the NMJ of the de novo limb muscles at stage 53. However, another (and we think more likely) explanation is that the decreased EMG response to d-tubocurarine is due to the drug acting centrally in diminishing the premotor activation of limb motoneurons during fictive axial swimming in these semi-isolated (CNSaxial/appendicular muscle) preparations. In support of this idea, we provide new immuno-anatomical labeling evidence for the presence of VAChT co-localized with synapsin around limb motoneuron cell bodies, indicating the presence of central cholinergic synapses on these neurons. That these synapses are involved in driving the limb MNs was confirmed with calcium imaging experiments on isolated CNS preparations demonstrating that their activation is substantially decreased during fictive swimming in the presence of d-tubocurarine. These additional results, which are illustrated in Figure 4—figure supplement 1 and briefly described in subsection “Locomotor-related activation of hindlimb muscles by non-cholinergic motoneurons” are therefore consistent with our study’s main conclusion that a switch in neurotransmitter phenotype of limb motoneurons occurs during early metamorphosis.

Finally, applying glutamatergic antagonists to the limb buds is certainly a potentially instructive approach and one that we had already attempted. Unfortunately, due the extremely small size of the limb buds in early stage animals (50-53), injecting drugs without destroying bud neuromuscular tissue and while maintaining a muscle recording electrode in place was found to be technically impossible.

2) Figure 5C,D: The EMG activity recorded at stage 53 is very weak and this may indicate that the neuromuscular junction is not fully developed.The authors should provide electrophysiological traces showing the effects of glutamatergic antagonists. Also, the weak EMG activity recorded at stage 53 might be due to the absence of activation of NMDA receptors. I suggest to repeat these experiments using a Mg-free extracellular solution.

The effects of glutamate antagonists on EMG responses to nerve stimulation are now shown in new Figure 6. A series of experiments was also done in saline containing half the normal MgCl2 concentration. As implied by the reviewer, the EMG response at stage 53 was increased (an illustration of the response of a stage 53 preparation in new Figure 6 (A, middle trace) is now taken from these experiments), but the effect of the glutamatergic antagonists (about a half-reduction) remained similar to that under normal saline.

3) Overall, it is not clear if limb motor neurons switch the transmitter phenotype or if they use both glutamate and acetylcholine as co-transmitters with a dominance of the one or the other at different developmental stages. In this regard, the authors could test if limb motor neurons co-express transporters for glutamate and acetylcholine in different proportions at different stages.

The expression of the vesicular transporter for glutamate VGluT1 in motor axons at early stages is now illustrated for a stage 53 tadpole in new Figure 5C. In addition, we provide as Figure 5—figure supplement 1 an illustration of VGluT1 immuno-detection in limb motoneuron cell bodies at stage 52, when ChAT and VAChT are not expressed. However, the expression of this transporter does not seem to persist at later stages, thus paralleling the lack of expression of NR1 and NR2b receptor subunits in the limb muscles.

Reviewer #3:

The central thesis of this paper is that motoneurons transiently release glutamate instead of acetylcholine at the neuromuscular junction of developing hindlimb muscles at the onset of metamorphosis. The main evidence in support of this conclusion comes from the immunocytochemistry showing that appendicular motoneurons do not express either ChAT or VAChT immunoreactivity and exhibit very little ChAT mRNA at stage 53. Moreover, at stage 52, the developing limb muscle has very low levels of α-bungarotoxin but does express NMDA receptor subunits some of which are co-localized with neuronal synaptophysin (although many are not).Unfortunately, the physiological experiments do not completely support the anatomy. For example, at stage 53 bath application of d-tubocurarine greatly reduces the amplitude of the rhythmic EMG signals from the limb bud during locomotor episodes despite the apparent absence of acetylcholine receptors in the muscle and cholinergic markers in limb motoneurons. The authors do not explain why the EMG is reduced at this age. Similarly, in the nerve/limb preparation illustrated in Figure 5, bath application of CNQX/AP5 only reduces the stimulus-evoked EMG by 50%. Why is the block incomplete? Again, this is not discussed by the authors. The authors showed only the presence of NMDA receptor subunits in the muscle but did not show the effect of NMDA antagonists alone on the evoked EMG to validate their presence.

As explained in our responses to similar comments made by reviewers 1 and 2, the experiments in Figure 4 were performed on semi-isolated CNS/muscle preparations where EMG activity in both the axial and limb muscles was driven spontaneously by the spinal locomotor network. We now provide new anatomical and calcium imaging evidence (Figure 4—figure supplement 1; subsection “Locomotor-related activation of hindlimb muscles by non-cholinergic mot”) that cholinergic synapses are present on limb motoneuron cell bodies and are substantially involved in the excitation of these cells during centrally-generated swim episodes. Thus, it is likely that the diminution of the limb EMG strength evident in Figure 4B resulted from a direct central effect of dtubocurarine on limb motoneuron excitation rather than at the peripheral neuromuscular junction.

On the other hand, the use of glutamatergic antagonists on the semi-intact preparation totally abolishes fictive swimming and so an equivalent EMG analysis could not be done with these antagonists. However, both cholinergic and glutamatergic sets of antagonists could be used on the isolated motor nerve-hindlimb preparation with nerve stimulation (old Figure 5, new Figure 6). The lack of a complete block by glutamatergic antagonists at stage 53 (Figure 6B, C) has several possible explanations. The first and simplest is that glutamate antagonists are not fully effective in the amphibian (especially in such an immature neuromuscular system) and cause a partial blockade only, as already reported (e.g., work from Robitaille’s group; Fu et al.,; Jamieson…). Another explanation is that glutamate is not the only neurotransmitter used at the limb neuromuscular junctions in these early stages. In relation to this possibility, we did an additional series of experiments using other neurotransmission blockers, but no clear results emerged (for example, picrotoxin was used to block chloride channels associated with GABA and Glycine receptors, on the basis that GABA has been found in immature *Xenopus* axial motoneurons in culture; e.g., work from N. Spitzer’s group). In addition, combining antagonists (notably, d-tubocurarine + glutamate antagonists) never led to a full blockade of stimuli-evoked EMG responses. We chose not to report these additional experiments in the revised manuscript to avoid over-complicating the main message and especially since it seems more generally that glutamate antagonists are unable to completely block glutamate-mediated transmission in amphibians.

In both experiments, it would have been useful to combine the glutamatergic and cholinergic antagonists to see if this produced a complete blockade of the muscle electrical activity.Finally, in Figure 5D, the results for control and d-tubocurarine are shown, but it would also be of relevance to show a recording for the glutamate antagonists. Here, the number of experiments is missing, and no p-values are reported.

As explained in our previous response, additional experiments in which we applied the glutamatergic and cholinergic antagonists in combination still failed to produce a complete blockade of stimulus evoked EMG responses. In any case, because usually neither drug could be fully washed out in our previous experiments, the first drug was still present if a second was applied sequentially. We also did control experiments where only one drug was applied, and the effects were similar to those obtained with two antagonists applied in combination. This latter observation is not illustrated but is stated in the Materials and methods section. The effects of glutamatergic antagonists are now illustrated for both early and late stages in new Figure 6. The numbers of experiments and p values are also now given in the corresponding text (subsection “Developmental switch in hindlimb neuromuscular transmission”).

The intracellular recordings of the locomotor activity illustrated in Figure 3E are puzzling. The recordings reveal rhythmic hyperpolarizing potentials and the cells do not fire. Was this true of all of the motoneurons that were recorded? If so, how are the muscles activated if the motoneurons do not spike? For the 11 recorded motoneurons, the authors should indicate from how many preparations they were obtained, and if the motoneurons were only identified by their labeling or whether they tested for an antidromic response. If it is the former, did the authors ever use a second dye applied intracellularly to confirm that the labelled motoneuron was indeed the one that was recorded from.At the earliest stages of limb formation, it was stated that motoneurons were small cells. However, the resistance of the cell shown in Figure 3C was between 3 to 5 MOhm suggesting either a very leaky cell or a large one. What was the input impedance of the other motoneurons?

The absence of firing in our recorded motoneurons was due to the internal solution that we generally used in our patch-clamp electrodes. We have performed a new set of experiments in which the patch pipette solution was changed for a high chloride solution, as usually used for recording immature motoneurons (see Akerman and Cline, now cited in the Materials and methods section). In this latter condition, neurons were able to fire action potentials spontaneously or in response either to current injection or spontaneous CPG activity. These experiments are illustrated in Figure 3C,D and F, and both the Results section and Materials and methods section have been updated accordingly.

Recorded motoneurons were indeed identified according to their labeling from hindlimb muscles, although fluorescent dye was also sometimes added in the patch pipette for later verification. This is now stated in the text and illustrated in Figure 3B. The number of preparations in each case is indicated in the Materials and methods section.

Early stage appendicular motoneurons have somata of 15-20 µm and were classified as 'small' neurons compared to the large axial motoneuron cell bodies (that are about three-fold larger than those of limb motoneurons). We are fairly confident that the intracellular recordings reported in this study were not ‘leaky’ (in fact, leaky cells were excluded from analysis), but since we have no comparison with other studies in this developmental range, we cannot draw any conclusions about the significance of the resistance values observed.

As already alluded to, a major concern is the lack of statistical analysis and an indication of the number of experiments used to generate the quantitative data. For example, in subsection “Delayed cholinergic transmitter phenotype expression in the developing appendicular motor Column” there is no statistical comparison between the fluorescence levels at the different ages nor is the number of experiments shown. Similar problems apply to the quantification in other parts of the paper. The authors often refer to the mean and the SE. Do they mean the standard error of the mean (S.E.M). Why was the standard error of the mean rather than the standard deviation used to illustrate the variability?

A statistical comparison was not reported for the fluorescence levels because we doubted the validity of this type of analysis. However, we have changed the way the fluorescence data are displayed, and statistics are also given. Note that this analysis has been redone according to the following comment.

The quantification of the fluorescence in Figure 2C was confusing. In the text, it states that ChAT and VAChT were not immuno-detected in appendicular MNs until stage 55. The figures in Figure 2A confirm this. However, in the plot of 2C (52-54), which shows the AP/AX fluorescence ratio, the values range from ~0-14% for ChAT. It is difficult to see how these values are generated if AP is zero. The authors should indicate how the ratio was calculated.

Cholinergic terminals are widely present in the spinal cord, including around limb motoneuron somata (now illustrated in Figure 4—figure supplement 1), which can sometimes affect the fluorescence ratio. We have redone the analysis in choosing as background a region without motoneurons but enriched in cholinergic terminals. This is now stated in the Materials and methods section and we are grateful to the reviewer for pointing out this analytical bias. The results of this analysis are presented differently from the original Figure 3C: we now show ∆F/F for axial and appendicular motoneurons separately, which also allowed reporting the statistics.

Also, despite an increase in the ratio at later stages, it is still about 20% of the axial fluorescence (when appendicular MNs are supposed to be bigger). This implies that the majority of appendicular motoneurons are still not cholinergic. It would be helpful to show that once the metamorphosis is complete, this ratio is 100% thereby validating the measure.

Our evidence indicates that all limb motoneurons become cholinergic from stage 55 onwards. However, axial motoneurons always displayed much more fluorescence than limb motoneurons, and this is even more evident in the new data representation in Figure 2C. Even in juvenile adults, i.e. after metamorphosis completion, the ratio remains largely in favor of axial motoneurons. Presumably this indicates a higher concentration of cholinergic proteins in axial motoneurons, but we do not know why this is so.

*I found the in-situ hybridization results shown in Figure 2D, difficult to interpret. It is difficult to know the extent of the ChAT ISH labelling in these Figures For example, the extent of the Axial motor column in D 53 looks much more limited than the ChAT labelling in A 53. Furthermore, it is not clear that appendicular MNs at stage 53 are negative. The background is very high. Sections with dextran labelling would have made clear where the motoneurons are located. This figure does not agree with the statement that ISH is more sensitive than immunodetection. How many times was this experiment performed?*

Islet1 in situ hybridization has been performed on the same preparations as ChAT hybridization. We now also illustrate the results for Islet1 (in Figure 3—figure supplement 1) in order to compare the location and extent of the appendicular motor column at key developmental stages. Whereas limb motoneurons were always Islet1 mRNA-positive, ChAT mRNAs were never detected at early stages (stage 51 now illustrated in Figure 2D). New images are displayed in which the axial motor column is clearly visible in all hybridizations, and the number of experiments has been added in the text (subsection “Delayed cholinergic transmitter phenotype expression in the developing appendicular motor column”).

mRNAs were detected with in situ hybridization before their translation into proteins. Thus, at pivotal stages 53-54, a light ChAT mRNA labeling could be observed in some preparations whereas ChAT proteins were never immuno-detected. This again is consistent with a delayed acquisition of the cholinergic phenotype by limb motoneurons.

[Editors' note: further revisions were requested prior to acceptance, as described below.]

The manuscript has been improved and reviewers 1 and 2 were satisfied with your amendments, but there are some remaining issues raised by reviewer 3 that remain to be addressed, indicated in the appended reviews. The other reviewers and I in consultation now agree on the need for these revisions:1) The verity of the VGluT1 localization, in synaptic terminals rather than in cell bodies; the predicted restriction of vGluT1 to motor neuron terminals and colocalization with synaptic proteins, and mRNA localization in motor neuron cell bodies; and a comparison to IHC of vGluT2. These aspects were deemed essential to your next revision.

A major remaining concern of reviewer 3 was our finding of VGluT1 localization in the immature limb motoneuron cell bodies, where he/she argued (along with two reference citations) that VGluT1 should be found exclusively in axon terminals, in the vicinity of synaptic contacts. However, we respectfully point out that there is evidence to indicate otherwise. Most previous studies on the subject (including the two reports cited by the reviewer) investigated glutamatergic projections onto non-glutamatergic neurons and accordingly, a strong concentration of VGluT1 immunoreactivity was detected in the vicinity of the latters' cell bodies, but which themselves unsurprisingly were nonreactive. In contrast, other studies on glutamatergic signaling in various brain and spinal structures (e.g., Malet and Brumovsky, 2015; Melo et al., 2013; Nakamura et al., 2005; Yang et al., 2014) have reported the expression of VGluT1 both within the somata of glutamatergic neurons, as well as along their axons. Although these studies were on rodents, and we have not found previous reports of similar findings in amphibians, the results do provide a clear precedence for our observation of cell body and axonal VGluT1 expression in immature *Xenopus* limb motoneurons.

Nevertheless, we performed a new set of immunolabeling experiments using a different VGluT1directed antibody (from mouse, instead of the guinea pig antibody originally used.). Here again, we obtained a very similar labeling in early limb motoneuron cell bodies (now illustrated in new Figure 5—figure supplement 1A) as with the guinea pig antibody. Additionally, VGluT1 expression was found along MN axons, as well as co-localized with synaptophysin and NR1 immuno-labeling (new Figure 5—figure supplement 1 B1) as would be expected with a functional glutamatergic NMJ. As also requested by the reviewer, we now illustrate that VGluT1 labeling is lost in stages older than 54 (Figure 5—figure supplement 1B2), when glutamatergic signaling disappears.

Unfortunately, after many attempts with various protocols (including that published by Morona et al., 2017 and even using their primer, which they kindly sent to us) we failed to produce VGluT hybridization, which if successful would have obviously reinforced our immunohistological observations. However, we believe that (1), finding very similar results with VGluT1directed antibodies originating from two different mammalian sources, (2), the general endorsement from the literature, and (3), the combination of the experimental data provided by our present study support the conclusion that before becoming cholinergic, de novo limb motoneurons at premetamorphosis initially use glutamate as a major transmitter at the NMJ.

The impetus for our VGluT1 immunolabeling experiments was based on a previous report (Gleason et al., Gene Expression Patterns, 2003) demonstrating the transporter's presence in the early developing *Xenopus laevis* spinal cord. Although our experiments using mammalian VGluT1 antibodies gave positive results, this of course doesn't mean that the *Xenopus* and guinea pig or mouse VGluT1 isoforms are identical. We also tested for VGluT2 immunolabeling (again using a mammalian-targeted antibody) in spinal cord cross sections and this failed completely. In this context, however, it is relevant that according to a phylogenetic analysis by Villar-Cervino et al., (2010), vesicular glutamate transporters in *Xenopus* are thought to exist as two related subtypes (a and b) of VGluT1, rather than as homologues of the distinct VGluT1 and 2 isoforms. In any case, we should emphasize that the aim of our VGluT immunolabeling experiments was to seek supporting evidence for (or against) glutamatergic synaptic signaling playing a role in the early developing limb system, rather than attempting to identify the specific transporter isoforms that might be involved.

The next two sets of comments could be dealt with in the text of the Discussion section:2) incomplete blockade of evoked muscle action by glutamate antagonists, that could be explained by the low concentration of glutamate antagonists used;

A new paragraph has been added to the Discussion section where we now address the incomplete blockade of evoked EMG responses by glutamatergic antagonists. We have also rewritten much of the Discussion section’s first two paragraphs in order to avoid implying that glutamate is alone responsible for neurotransmission at the early limb NMJ.

3) physiology experiments showing that the reduction of the calcium transients generated during fictive swimming by tubocurarine may be due to blockade of GABA-A or glycinergic receptors rather than cholinergic receptors.

This interesting and plausible possibility is now mentioned in the Results section. Indeed, the possible contribution of a d-tubocurarine-sensitive, GABA-A receptor-mediated drive to limb motoneurons reinforces our hypothesis that in spontaneously active semi-intact preparations, d-tubocurarine causes a central blockade of motoneuron activation, rather than acting peripherally at the limb NMJ.

Reviewer #3:

In response to the reviewers' concerns the authors have attempted to demonstrate that motoneurons express VGluT1. Unfortunately, I did not find their presentation of the data convincing. For example, VGluT1 expression at stage 52 (Figure 5 supplement 1) is widespread throughout the ventral cord and appears, surprisingly, to label cell bodies. This is in contrast to numerous other immunocytochemical studies of VGluT1 labelling in the spinal cord that show the labelling is restricted to synaptic terminals. (Betley et al., 2009; Schultz et al., 2017). The authors should illustrate an image of the whole transverse section of the cord to show how widespread the labelling is. Was this antibody tested or has it been described in other papers to show that it is indeed targeting VGluT1? What experiments did the authors perform to examine the specificity of the antibody? The VGluT1 labelling shown in Figure 5B appears coextensive with the muscle nerve but there is no evidence of the more punctate labelling that characterizes the neuromuscular junction and is shown in Figure 5A (α-bungarotoxin). Furthermore, the authors should show the VGluT1 labelling at stage 57 when glutamatergic transmission is absent. Finally, it was not clear if the authors had tried to label the cells with VGluT2 whose mRNA has been reported in some studies to be expressed in the neonatal mammalian motoneurons.A more convincing approach to demonstrate the presence of a vesicular protein in motoneurons would be to show that it co-localizes with a synaptic protein – such as synaptophysin at the neuromuscular junction or to demonstrate the presence of the mRNA in motoneuronal cell bodies.

As detailed in our response to editorial comment 1 above, our observation of a diffuse VGluT1 immuno-reactivity in the somata of early *Xenopus* limb motoneurons finds strong echoes in the literature. Similarly, the expression of VGluT1 along glutamatergic neuron axons was also reported in these studies, which also corresponds to our findings in immature limb motoneuron axons.

The new series of VGluT1 immuno-detection experiments using a different antibody (generated from mice) provided a test for antibody specificity. The finding of VGluT1 expression in MN somata was the same as with the original guinea pig antibody, and these data are now illustrated in new Figure 5—figure supplement 1A. In addition, combined VGluT1, synaptophysin and NR1 immuno-detection in early hindlimbs showed co-localized labeling, consistent with functional neuromuscular synapses (Figure 5—figure supplement 1B1). Finally, we provide new images illustrating the parallel loss of VGluT1 and NR1 immuno-detection in hindlimbs in stages older than 54 (Figure 5—figure supplement 1B2).

According to a phylogenetic analysis by Villar-Cervino et al., (2010) a homologue of the mammalian or fish VGluT2 gene doesn't not exist in *Xenopus*, and what is usually termed VGluT2 is in fact a subtype of the VGluT1 isoform due to the tetraploid nature of the animal. Contrary to what is observed in other species, the amino acid sequences differ very little between the two resulting proteins, thus corresponding to VGluT1a and VGluT1b. Although no distinction could be made in our experiments between these two VGluT1 subtypes, this would explain our failure to see any VGluT2 labeling using a mammalian antibody.

Frustratingly, our repeated attempts (and with different protocols) failed to achieve VGluT hybridization. However, as stated in our response to editorial comment 1, we believe that the combination of our existing immunolabeling and electrophysiological data provide convincing evidence that pre-metamorphic limb motoneurons initially use glutamate as a neurotransmitter before becoming cholinergic.